# Rejection-based choices discourage people from opting out of voting

Yi-Hsin Su [1] ✉ & Amitai Shenhav [1,2] ✉

Representing the will of voters is challenging when many opt out of indicating their preferences. Such opt-out behavior has been explained by voters lacking a preference and/or disliking their options. We provide evidence for a third account: people opt out of choosing between undesirable candidates because bad options are incongruent with their typical goal of selecting the best one. Using a voting task, we show across two lab-based studies that the tendency to opt out of choices between bad candidates is eliminated when participants are asked to reject the worst candidate. Leveraging our experimental findings, we simulate elections and show that rejection-based voting can produce election outcomes that are more representative of the preferences of the electorate. To validate this prediction, we conduct two Prolific surveys of self-reported US Independents before the 2024 US presidential election, and show that people are less likely to respond "undecided" when asked who they will vote against rather than who they will vote for. Our findings help understand when and how people vote, and how to better reveal the preferences of voters who know which candidates they like least but are unwilling to endorse the one they like most.

When seeking to represent the will of the people—whether it is the leaders or policies they most prefer—all democracies face the challenge of measuring the preferences of their voters. A fundamental barrier to representing these preferences is that many voters opt not to indicate a preference in a given election, either passively (e.g., not responding to a poll or not showing up to the ballot box)[1–5] or actively (e.g., abstaining or responding "undecided")[6,7]. Such opt-out behavior is observed in a substantial proportion of the electorate[4], one that is growing worldwide[8] and producing significant gaps in voter representation[9]. To better capture the will of the people, it is therefore critical to understand what motivates such opt-out behavior in the context of voting, and to identify ways of revealing the preferences of these voters.

One reason why people might opt out of indicating their preference in an election is if they would be equally satisfied with any of the outcomes (e.g., with either of two candidates winning)[10,11]. To the extent opt-out behavior is purely motivated by *indifference*, it can be viewed as having a negligible impact on voter representation since withholding a vote will, on net, have a similar impact as choosing based on equal preference (e.g., flipping a coin). For instance, if all non-voters felt the same way about Candidates A and B, then getting these indifferent voters out to vote wouldn't change the election outcome because both candidates would end up yielding the same number of additional votes.

Another reason why voters might opt out of voting is if they are unhappy with their options[10,11]. Unlike an indifference-based account, this *alienation*-based account proposes that people may indeed have a preference between the two candidates but feel that this is a "lose-lose" choice. To the extent voters opt out based on alienation rather than indifference, the failure to reveal their preferences can be highly consequential. For instance, a group of voters may choose not to vote because they dislike both Candidate A and Candidate B, but dislike B more than A (i.e., have a preference for A over B). Unlike for indifferent non-voters, getting these alienated non-voters to vote can have a material impact on the election outcome by shifting the vote count in favor of their preferred (less disliked) candidate (e.g., Candidate A in this case).

[1]Department of Psychology, University of California, Berkeley, Berkeley, CA, USA. [2]Helen Wills Neuroscience Institute, University of California, Berkeley, Berkeley, CA, USA. ✉e-mail: yi-hsin_su@berkeley.edu; amitai@berkeley.edu

Here, we design a voting choice task to test the role of indifference and alienation in driving decisions of who to vote for and/or whether to vote. We then test an alternative to the alienation-based account: that lose-lose choices engender opt-out behavior not because the candidates are bad but because being bad makes these options incongruent with the typical choice goal ("who would I vote for?"). Our work builds on recent theoretical and experimental findings in decision science, which show that participants weigh information about their options (e.g., consumer goods) differently depending on their choice goal[12–15]. When facing multiple options they like, participants choose faster when their goal is to select the best option rather than the worst one. When facing multiple options they dislike, they are faster to choose the worst one than the best one. These findings have been accounted for by models that characterize decision-making as a process of evidence accumulation towards a decision threshold, whereby evidence congruent with one's decision goal (choose worst vs. best) reaches threshold fastest[14–16].

Our current work builds on these previous findings in two important ways. First, by testing whether goal-driven reversals in choice dynamics generalize from consumer choice to voting decisions (e.g., resulting in faster choices when selecting the better of two good candidates or choosing the worse of two bad candidates). Second, by testing whether these distinct choice goals also invert patterns of decision avoidance (i.e., whether to choose or opt-out) depending on the quality of one's options. If decisions about *whether* to choose are driven by goal congruency in the same way as decisions about which option to choose, then decision-makers would be expected to opt-out more or less depending on whether the quality of their option set aligns with their choice goal. In the context of voting decisions, this would predict participants opting not to vote when being asked to choose the better of two candidates they dislike, but not when asked to reject the worse of the two.

Across two experiments, we show that inverting the goal to have participants choose which candidate they would vote against (cf. anti-candidate voting[17] or negative voting[18–23]) leads to a suppression of alienation-based opt-out behavior—participants were now much less likely to opt out of those lose-lose choices than they would with the typical goal. We then test a direct implication of our voting congruency account for real-world poll outcomes, providing an experimental test of the influence of positive versus negative voting on people's stated preferences. Across two polls of the US presidential election, we show that poll responders are significantly more likely to reveal their preference between the major candidates if they are asked which candidate they would vote against rather than (as is typical) which candidate they would vote for.

## Results

### A laboratory-based measure of voter opt-out behavior

We recruited 100 participants to perform a voting task ($N = 91$ in the final sample; see "Methods" for inclusion and exclusion criteria and Supplementary Table 1 for participant demographics). In this task, participants first identified their positions on a series of political issues (e.g., abortion rights, gun policies, etc.) and indicated how important each issue is to them (Fig. 1a; for details of issues and item-level analyses, see "Methods" and Supplementary Tables 7 and 8). Based on this information, we were able to synthesize candidates—each characterized by their position on two issues (Fig. 1b)—who we predicted would be more or less desirable to that participant based on how aligned the candidates' positions are with the participant's own, and how important those issues are to the participant (Supplementary Fig. 1a, see

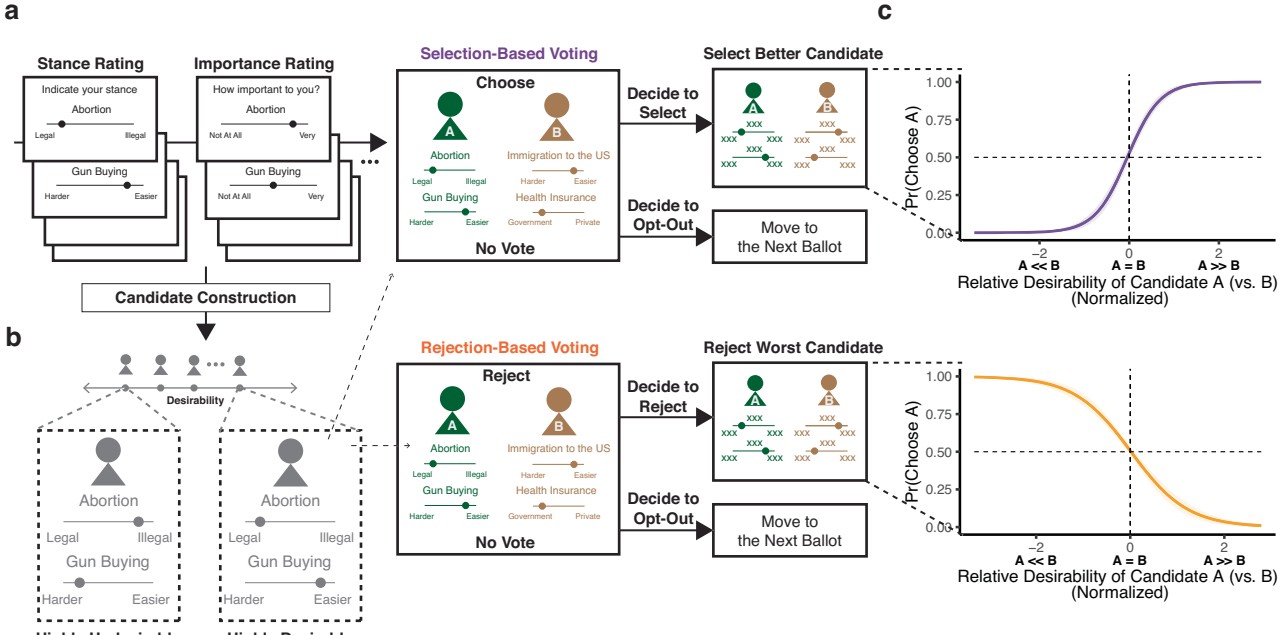

**Fig. 1 | Voting task. a** Participants first viewed a series of 13 political issues, and for each issue, participants indicated their positions and the importance of the issue to them. They then viewed a series of 100 ballots involving pairs of hypothetical candidates. Depending on their randomly assigned group, participants were asked to either vote for the better candidate (Selection-Based Voting, top) or vote against the worse candidate (Rejection-Based Voting, bottom). For each ballot, all participants first had the option of either (1) going on to vote on the ballot (select or reject from the pair) or (2) indicating a "no vote" (opting out). If they opted out, they would move directly to the next ballot. **b** We used a given participant's issue stances, weighted by their importance to the participant, to synthesize a wide array of candidates who varied in their alignment with that participant's views, from those who are well-aligned (highly desirable) to completely misaligned (highly undesirable). Candidates were paired together on ballots so as to vary the difference in desirability between them (relative desirability) and how desirable the two are on average (overall desirability). **c** The more desirable one of the candidates was than the other (e.g., Candidate A on the left side vs. Candidate B on the right side), the more likely participants were to choose them in the Selection condition (top, $N = 44$ for Study 1), and the more likely they were to choose the opposite side in the Rejection condition (bottom, $N = 47$ for Study 1). Pr: Probability. Shaded error bars show 95% confidence intervals.

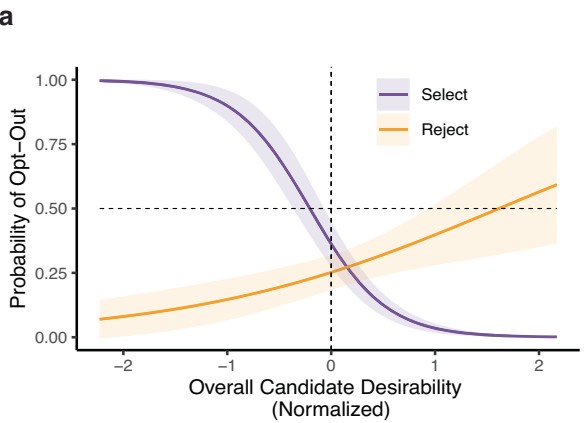

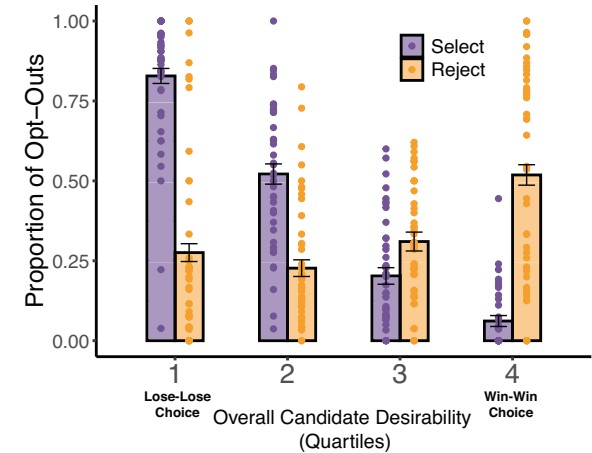

**Fig. 2 | Rejection-based voting selectively reduces opting out with undesirable candidates. a** Participants in Study 1 who were asked to vote for the better candidate (Selection group, purple, $N = 44$) were much more likely to opt out the more undesirable the candidates were overall. Participants who were asked to vote against the lesser candidate (Rejection group, orange, $N = 47$) showed the opposite trend—the more desirable the candidates were, the more likely to opt-out. Shaded error bars show 95% confidence intervals. **b** As a result, when participants were faced with lose-lose choices (lowest quartiles of candidate desirability; number of trials for each quartile: 1100 for the Selection condition, 1175 for the Rejection condition), they were three times (82.8% vs. 27.6%) more likely to opt out of voting to select than reject. Error bars show 95% confidence intervals.

"Methods"). For instance, for a participant who indicated strong support for increased legalization of abortion and rated this issue as very important, we could synthesize candidates who share this view (highly desirable), or who have a diametrically opposed position (highly undesirable), or who carry any level of desirability in between (e.g., because they hold intermediate positions on this issue, or because the issues they align with the participant on are of more moderate importance to that participant).

Participants viewed a series of ballots, each consisting of two of these hypothetical candidates. Across these ballots, we systematically varied (a) how (un-)desirable the two candidates are on average (overall desirability) and (b) how much more desirable one candidate is than the other (relative desirability) (Supplementary Fig. 1b). Critically, for each ballot, participants were first given the opportunity to opt out of choosing a candidate. If they chose this "No Vote" option, they would move on to the next ballot. If they chose to vote, they would move on to a second stage of the trial, where they would select one of the two candidates. Notably, similar to real-world voting, opting out carried no penalty and was time-saving, offering the opportunity to complete the experiment sooner.

Participants were randomly assigned to one of two conditions (Fig. 1a). Participants in both conditions performed identical voting tasks, with one exception: when choosing between hypothetical candidate pairs, one group was asked to select the candidate they preferred more (Selection condition), whereas the other was asked to reject the candidate they preferred less (Rejection condition).

To validate our approach to varying the desirability of hypothetical candidates based on each participant's positions, we first examined trials in which participants opted to vote. We found that both groups made choices consistent with what would be predicted based on the projected desirability of the two candidates and based on their respective voting goals. Participants in the Selection group were more likely (79.0% of all ballots) to choose the candidate more aligned with their policy views (more desirable); participants in the Rejection group were more likely (72.7% of all ballots) to reject the candidate less aligned with their policy views (less desirable). For both groups, participants were more likely to choose the candidate that better aligned with their choice goals (more desirable candidate in the Selection group; less desirable candidate in the Rejection group) as candidate desirability was more distinct from one another (higher relative desirability; Selection: log odds = 2.75, $z = 13.9$, $p < 0.001$, 95%

confidence interval (CI) = [2.36, 3.14], Supplementary Table 9 and Fig. 1c, top; Rejection: log odds = −1.67, $z = −11.1$, $p < 0.001$, CI = [−1.97, −1.38], Supplementary Table 11 and Fig. 1c, bottom). These patterns were also reflected in the speed with which participants made their decisions: faster when the candidates were predicted to be more dissimilar from one another (i.e., one was much more desirable than the other) and slower when they were predicted to be more similar to each other (Selection: $\beta = −0.04$, $t(45.0) = −5.61$, $p < 0.001$, CI = [−0.05, −0.02], Supplementary Table 10 and Supplementary Fig. 2a; Rejection: $\beta = −0.02$, $t(49.1) = −2.76$, $p = 0.008$, CI = [−0.03, −0.00], Supplementary Table 12 and Supplementary Fig. 2b).

## Vote participation is lowest for lose-lose choices, but is restored by rejection-based voting

Thus, when participants chose to vote, their votes were aligned with their preferences. For any given ballot, though, participants could also choose not to vote, and did so frequently. In the Selection condition, we found that participants opted out of voting on 40.3% of ballots and that the likelihood of opting out could be predicted by two main characteristics of the ballot. First, and perhaps most intuitively, participants were more likely to opt out of voting when the candidates were similarly desirable (lower relative desirability, log odds = −0.92, $z = −9.05$, $p < 0.001$, CI = [−1.12, −0.72], Supplementary Table 15 and Supplementary Fig. 2c), consistent with the indifference-based account. However, controlling for this relative desirability effect, we also found that opt-out decisions were sensitive to the overall desirability of their candidate options—these participants were much more likely to opt out of voting the lower the desirability of these candidates (lower overall desirability, log odds = −2.72, $z = −13.1$, $p < 0.001$, CI = [−3.13, −2.31], Supplementary Table 15 and Fig. 2a, purple line), consistent with the alienation-based account. In fact, the undesirability of one's options (i.e., having to face a "lose-lose" choice) emerged as by far the strongest predictor of opting out, with participants choosing to opt out of voting 82.8% (CI = [80.5%, 85.2%]) of the time when selecting between the bottom quartile of overall desirability (the most undesirable candidate pairs) compared to an opt-out rate of 6.2% (CI = [4.4%, 7.9%]) when selecting between the top quartile (Fig. 2b, purple bars). This pattern also contrasts notably with factors contributing to choices of which candidate to select, which were driven primarily by relative rather than overall desirability (Supplementary Table 24 and Supplementary Fig. 3).

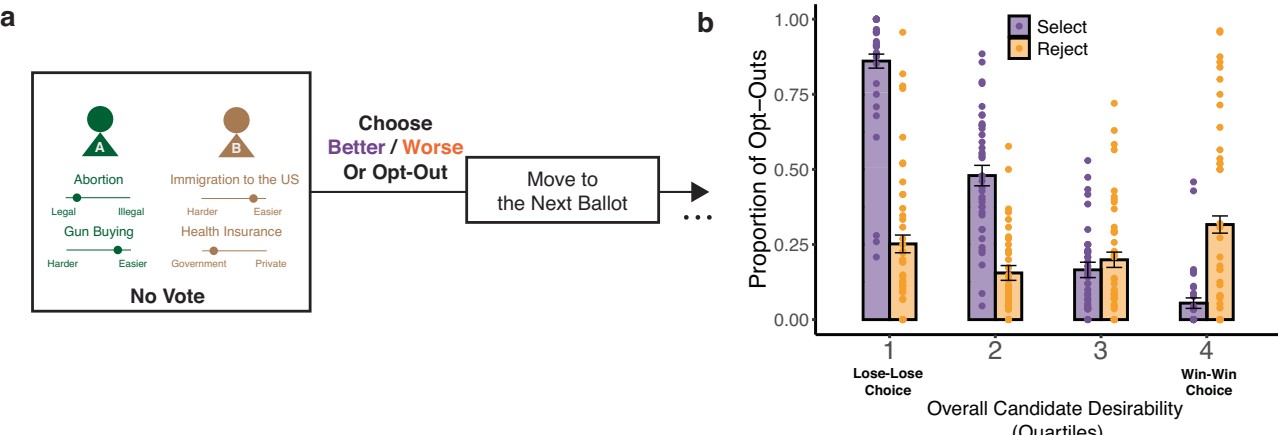

**Fig. 3 | Rejection-based reductions in opt-out behavior extend to compulsory choices. a** Participants in Study 2 performed the same task as shown in Fig. 1, but instead of making a series of binary choices (vote vs. no-vote, then Candidate A vs. B), they made a single choice among the three core options (Candidate A, Candidate B, and no-vote). **b** Mirroring the patterns we observed in Study 1 (Fig. 2b), participants in the Selection group (purple bars, $N = 39$; number of trials for each quartile: 975) were again much more likely to opt out of this choice than participants in the Rejection group (orange bars, $N = 43$; number of trials for each quartile: 1075), particularly true for "lose-lose" choices (86.0% vs. 25.2%). Error bars show 95% confidence intervals.

Opt-out behavior was markedly different for participants in the Rejection group. Overall, Rejection participants were less likely to opt out of voting than Selection participants (Rejection: 33.3% vs. Selection: 40.3%, $p = 0.013$, Brunner–Munzel test statistic (degree of freedom, df) = 2.54 (85.7), two-sided), but where they differed most was in how these opt-out choices varied with the overall desirability of the candidate options. Whereas Selection participants showed a steep increase in opting out as the desirability of their options decreased (Fig. 2a, purple line), Rejection participants did not (log odds $_{goal \times overall\ desirability} = 1.71$, $z = 10.2$, $p < 0.001$, CI = [1.38, 2.04], Supplementary Table 17 and Fig. 2a, orange line). Compared to the 82.8% (CI = [80.5%, 85.2%]) opt-out rate for the bottom quartile of ballots when selecting the best candidate, participants who rejected the worst candidate only opted out of 27.6% (CI = [24.8%, 30.3%]) of these ballots (Fig. 2b).

Interestingly, rejection instead led to a modest trend in the direction opposite of what was seen for selection, with Rejection participants opting out more for most desirable candidates (log odds = 0.71, $z = 2.65$, $p = 0.008$, CI = [0.19, 1.24], Supplementary Table 16 and Fig. 2a, orange line). This reversal can reflect participants having difficulty deciding which to reject when both candidates are highly desirable. Consistent with this interpretation, when participants opted to choose between the candidates we found that their patterns of decision times reversed depending on their choice goal, in a manner consistent with studies of consumer choice[14–16]: participants chose fastest when selecting the better of two highly desirable candidates ($\beta = -0.04$, $t(38.0) = -4.47$, $p < 0.001$, CI = [−0.06, −0.02], Supplementary Table 10 and Supplementary Fig. 4a) and when rejecting the lesser of two highly undesirable candidates ($\beta = 0.02$, $t(39.1) = 2.06$, $p = 0.046$, CI = [+0.00, 0.03], Supplementary Fig. 4a; interaction: $\beta_{goal \times overall\ desirability} = 0.03$, $t(83.0) = 4.62$, $p < 0.001$, CI = [0.02, 0.04], Supplementary Table 14 and Supplementary Fig. 4a). Collectively, these findings confirm our prediction that candidate rejection can decrease tendencies to opt out of voting for undesirable candidates by making it easier to choose between these candidates.

### The benefits of rejection-based voting extend to abstention from compulsory voting

Findings from this first study suggest that framing an election in terms of candidate rejection has the potential to reveal otherwise hidden voter preferences by reducing the likelihood of an individual opting

out of a vote. However, even when one chooses to participate in an election (including in cases where this is compulsory), it is possible to abstain from voting by actively indicating one's preference for a null alternative (e.g., "abstain," "no vote," or "uncommitted"). The same applies to situations where a person is actively polled for their preference in advance of an election, in which case they often have the option to decline from committing to a single candidate by indicating that they remain undecided. To test whether the benefits of rejection-based voting carry over to forced-choice contexts like these, we recruited a separate group of 100 participants to replicate Study 1's findings in a modified version of our voting task (Study 2, Fig. 3a; see "Methods: Statistical Analysis" for power analyses). Study 2 was identical to Study 1 except that rather than being able to make a separate opt-out decision prior to selecting between the candidates, participants instead made a single choice with each ballot, with their options consisting of the two candidates and a third option allowing them to abstain from voting on that ballot ("No Vote").

Mirroring our first study, we found that participants faced with these three options and charged with selecting the best candidate ($N = 39$) were most likely to select the no-vote option due to alienation (low overall desirability, log odds = −2.89, $z = −13.0$, $p < 0.001$, CI = [−3.33, −2.46], Supplementary Table 15). Faced with the same three options, participants who were charged with rejecting the worst candidate ($N = 43$) were, on the whole, significantly less likely to opt out of voting (23.1% vs. 39.0%, $p < 0.001$, Brunner–Munzel test statistic (df) = 6.68 (77.8), two-sided). Once again, the decrease in opt-out behavior was most pronounced when making lose-lose choices (e.g., 25.2%, CI = [22.3%, 28.1%] vs. 86.0%, CI = [83.7%, 88.4%] for the lowest quartile of overall desirability; Fig. 3b) and was modestly reversed when making win-win choices (e.g., 31.6%, CI = [28.8%, 34.5%] vs. 5.5%, CI = [3.7%, 7.2%] for the highest quartile; Fig. 3b). We found a similar reversal in decision time when participants opted to vote (interaction: $\beta_{goal \times overall\ desirability} = 0.03$, $t(104) = 6.07$, $p < 0.001$, CI = [0.02, 0.04], Supplementary Table 14 and Supplementary Fig. 4b), again indicating that rejection became more difficult as candidate desirability increased.

### Rejection-based voting enhances voter representation by mobilizing alienated voters: evidence from population-level simulations

Across these two studies, we see at the individual level that candidate desirability exerts an influence on voting behavior in two ways. The

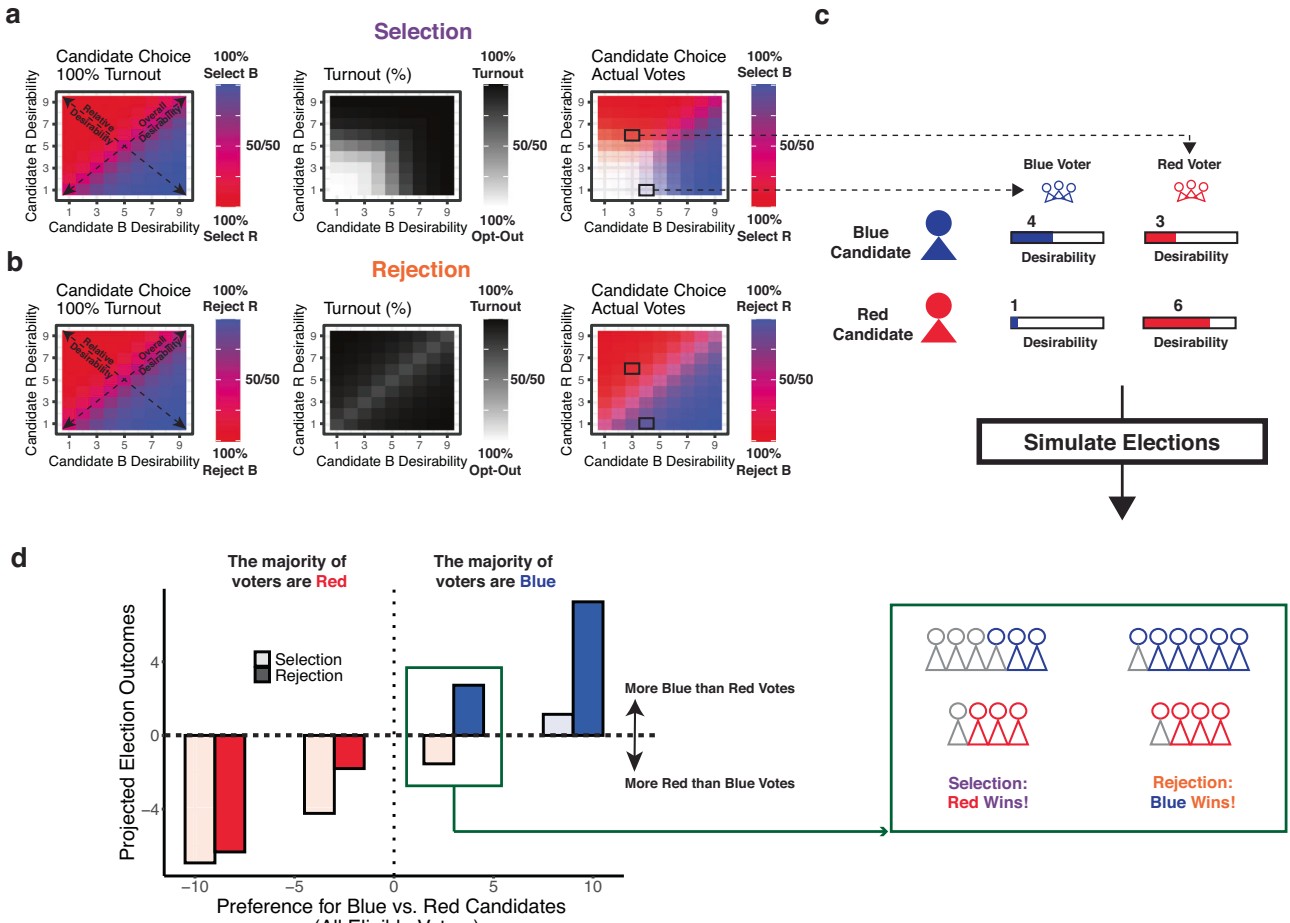

**Fig. 4 | Simulations of selection- and rejection-based elections. a** We simulated populations of voters based on patterns of behavior observed across Study 2 participants (see Supplementary Fig. 5 for Study 1), separately for Selection-based voting (**a**) and Rejection-based voting (**b**). **Left:** Under conditions where all voters cast a ballot for one of two candidates, these simulations predict that participants will be more likely to select a candidate the more desirable they are relative to the other candidate (blue-red gradient). **Middle:** When an opt-out option is introduced, participants increasingly opt out of voting, the more undesirable the candidates are (black-white gradient). **Right:** As a result, these "lose-lose" voters will be less represented in the election (white area), leading to election outcomes that are determined both by the relative desirability of the candidates (primarily determining choice if voters vote) and their overall desirability (primarily determining whether a voter will vote). **b** With rejection-based voting, opt-out behavior (middle)

is much less determined by overall desirability and more by how similar the candidates are (i.e., voter indifference between the outcomes). Voters are, therefore, well-represented across levels of overall candidate desirability. **c** We simulated conditions in which two sets of voters favor opposing candidates by the same margin (e.g., 3 points difference in desirability), but Red voters like both candidates more overall (6 vs. 3 for Red vs. Blue Candidate), whereas Blue voters like both candidates less (1 vs. 4). **d** Under these conditions, in a traditional (selection-based) election, the Red candidate can win the election even if the Blue voters are in the majority (lighter bars), given the unequal levels of opt-out behavior between the two sets of voters. In a rejection-based election (darker bars), election outcomes are more representative of the population preferences (reflected in more symmetric bars around the horizontal midpoint).

relative desirability of the candidates (the extent to which one is preferred to the other) determines which candidate would be selected if they were to vote. By contrast, the overall desirability of the candidates (the extent to which the candidates are seen as good or bad options) determines whether a vote is cast for *any* of the candidates or whether the voter opts out of making a choice. Given that voter preferences at the population level are determined only by the votes that are cast, one implication of this systematic source of voter opt-outs is that it could produce preference measures (e.g., election or poll outcomes) that deviate from the preferences of the majority of eligible voters (i.e., from what would be expected based only on how much the population prefers one candidate over another). To explore this possibility, we used choice data from our voting task to simulate agents with varying preferences for two hypothetical candidates ("Red Candidate" vs. "Blue Candidate") and to project the likelihood that agents will vote for each of those candidates or opt out, under different voting conditions (Fig. 4 and Supplementary Fig. 5, see details in "Methods").

When simulating conditions in which all agents are forced to select which of the candidates they prefer (i.e., no possibility of opt-outs, or "full turnout"; Fig. 4a, left panel), we see that the likelihood of the Blue Candidate receiving a given agent's vote (the blue-red gradient) depends primarily on how much better that candidate is than the other candidate (relative desirability; deviation from the diagonal line), independent of how much the voter likes the Blue Candidate (overall desirability; deviation from the origin). If we simulate the same elections/polls in a world in which voter opt-outs are permissible (Fig. 4a, middle panel), we see that such opt-outs would occur disproportionately in cases where agents find both candidates unappealing (alienated voters; white areas in the bottom-left corner), even if one of these candidates is consistently less unappealing than the other.

These findings imply that, as a result of their increased opt-out rates, voters who dislike both candidates will have less influence on the ultimate preference measures (Fig. 4a, right panel). This can be demonstrated by simulating two hypothetical sets of voters who have

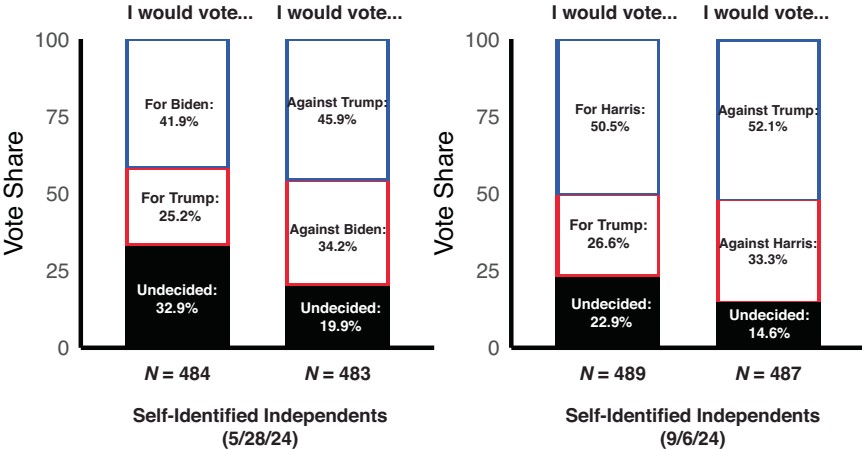

**Fig. 5 | Poll respondents are less likely to opt out of voting against (vs. voting for) a US presidential candidate.** In two preregistered surveys, self-identified Independents were randomly assigned to indicate whether they would vote for (Selection group) or vote against (Rejection group) one of the two major candidates in the 2024 US presidential election (survey data collected on Prolific: 05/28/2024 for Joe Biden vs. Donald Trump and 09/06/2024 for Kamala Harris vs. Donald Trump). Both groups were also given the option to indicate that they were undecided. Survey respondents were significantly more likely to respond 'undecided' when they would otherwise need to vote for a candidate (Study 3: 32.9%; Study 4: 22.9%) than if they would otherwise need to vote against a candidate (Study 3: 19.9%; Study 4: 14.6%).

opposite rankings of the two candidates (Fig. 4c): Blue Voters, who prefer the Blue Candidate over the Red Candidate by a 3-point margin; and Red Voters, who prefer the Red over the Blue Candidate by an equivalent (3-point) margin. The only difference between these two voter groups is that the Blue Voters like both candidates less (rating the Red Candidate as a 1 and the Blue Candidate as a 4), whereas the Red Voters like both candidates more (6 for Red vs. 3 for Blue). In a case like this, we find that even when the majority of the population are Blue Voters, because these voters are more prone to opt-out, the election/poll can result in a win by the Red Candidate (Fig. 4d, lighter bars). The same bias would hold in reverse if the majority of the population were Red voters who preferred both candidates less than Blue voters; in either case, the voters who view the candidates as more desirable would be predicted to benefit from an advantage in representation.

These cases of preference measures deviating from population preferences occur when simulating traditional, selection-based choice (i.e., choosing the better candidate). When we instead simulate voting behavior under rejection-based choice (Fig. 4b), we find that preference measures are much more faithful to the preferences of the population of eligible voters (Fig. 4b, right panel), such that the Blue versus Red Candidate is expected to win when Blue versus Red Voters are in the majority, largely irrespective of the overall desirability of the candidates (Fig. 4d, darker bars). This results from the diminished impact of overall desirability on opt-outs by individual voters and the increased turnout of alienated voters (Fig. 4b, middle panel). Consequently, an agent's likelihood of voting for the Blue Candidate (by rejecting the Red Candidate) is primarily determined by relative candidate desirability (Fig. 4b, right panel), mirroring choices in settings with full turnout (Fig. 4b, left panel).

**Rejection-based framing reduces opting out of candidate selection in real-world election surveys**

By systematically varying the desirability of candidates for individual participants, our two laboratory studies show that rejection-based voting (rejecting the worst candidate) can be effective at reducing a given person's tendency to opt out of voting for certain choices (those between undesirable candidates) and not others. To test whether our findings generalize to real-world voting decisions, we ran two pre-registered survey studies during the period of the 2024 US general elections at different time points (May 28th for Study 3 (https://osf.io/djy4h) and September 6th for Study 4 (https://osf.io/89e75); note that

the major candidates had changed over this period). Based on our simulations from Studies 1 and 2 (Fig. 4), we expected the selection vs. rejection framing to have the largest impact on opt-out behavior for those participants who tend to experience their candidate choice in the US elections as a lose-lose. Therefore, for both surveys, we recruited a thousand US participants on Prolific who self-identified as Independents—a group that during this period was typically most likely to hold negative views of both Democratic and Republican candidates[24]—to respond to a poll for the upcoming elections (final $N = 967$ for Study 3 and 976 for Study 4; see "Methods" for inclusion and exclusion criteria). Survey responders were randomly assigned to indicate which of the two major candidates at that time (Joe Biden vs. Donald Trump for Study 3; Kamala Harris vs. Donald Trump for Study 4) they intended to vote for (Selection-Based Polls) or which candidate they planned to vote against (Rejection-Based Polls). Critically, both conditions included an identical third option ("I'm undecided"), which could serve as an opt-out option for respondents who did not want to commit to either candidate. When responding to the traditional, selection-based poll format, a substantial proportion of respondents opted not to commit (Selection: 33% for Study 3 and 23% for Study 4). As predicted, we found that having respondents instead indicate who they would vote against resulted in a significant decrease in this opt-out-like behavior (Rejection: 20% for Study 3 and 15% for Study 4; Rejection < Selection, two-proportion one-sided $z$ test: $\chi^2 = 20.3$, df = 1, $p < 0.001$ for Study 3 and $\chi^2 = 10.6$, df = 1, $p < 0.001$ for Study 4; Supplementary Tables 18 and 19 and Fig. 5), regardless of the timepoint of the polls or the candidates involved. These findings do not change as we further include those who responded to the poll using the option "Prefer not to respond" ($N = 11$ for Study 3 and $N = 8$ for Study 4; see "Methods" for the poll design), potentially treating this option as another route to opt out of voting (Supplementary Tables 18 and 19).

## Discussion

The outcomes of any election hinge both on who people are likely to vote for and whether they are likely to vote. Our findings suggest that the answers to these two questions lie, to different degrees, in two elements of the options under consideration. Using a laboratory-based voting task, we show that the relative desirability of the candidates (the extent to which participants like one more than the other) primarily determines *which* candidate will be chosen if they opt to choose one of the candidates, whereas the overall (un-)desirability of the candidates

(how much the participant likes each one) primarily determines *whether* they opt to choose any of the candidates. Critically, we show that the influence of candidate desirability on whether a person chooses to vote depends heavily on how the voting decision is framed. People are more likely to opt out of choosing when they are asked which of two bad options to vote for, but not when they are asked which of those lose-lose options they would vote against. Our survey results demonstrate the same framing effects when poll respondents decide whether to choose a preferred candidate versus opting not to choose. Participants were more likely to opt out of indicating a preference between presidential candidates (were more likely to say they were undecided) when asked which candidate they would vote for compared to when they were asked which candidate they would vote against.

Past work suggests that voters may opt out of indicating a preference (either by withholding a vote or voting undecided) because they are indifferent (i.e., lack a clear preference between the candidates)[10,11] or ill-informed (i.e., lack sufficient information about them)[25,26]. We were able to show that, holding both of these factors constant, decisions about whether or not to vote are to a large degree driven by how desirable the candidates are in absolute terms (e.g., whether the participant views the choice as lose-lose rather than win-win). On its own, this finding argues for alienation-based accounts of voter abstention, which propose that voters may withhold a vote when they are unhappy with their options, a situation representative of recent elections in the United States[27,28] and elsewhere[22,23]. However, drawing on recent models of preference-based choice[14–16], we proposed and validated an alternative account: voter opt-outs are driven not by bad candidates but by candidates being incongruent with the selection-based goal. We show that so-called alienated voters are more willing to choose a candidate—both in simulated and real elections—if the typical decision frame is inverted, from selecting the better candidate to rejecting the lesser candidate.

Our experimental findings enabled us to simulate and generate testable predictions regarding the combined impact of relative and overall candidate desirability on measures of voter preference (e.g., elections and polls), which ultimately only reflect the preferences of those who opt to provide them. We show that, under a typical selection-based ("vote for") framing, such measures will be biased towards the preferences of voters who have a more positive overall impression of the candidates and will be less representative of voters who have a preference but choose not to share it because they are unwilling to indicate which candidate is the "lesser of the two evils." By contrast, rejection-based framing is more likely to reveal the preferences of these voters and, therefore, to provide a more representative sample of voter preferences in the wider electorate. Our survey results validate these predictions, showing that nearly 40% fewer self-identified Independents opted out of revealing their preference between the major presidential candidates when given a rejection-based rather than a selection-based poll. Together, these findings offer an alternative to common accounts of undecided voters as merely indecisive[29] or ill-informed[30], and instead suggest that many may be simply unwilling to positively endorse any of the candidates.

Our studies advance past research on voting behavior by experimentally testing the influence of a negative voting manipulation on real-world measures of voter preferences, using a randomized controlled design. Past empirical work on negative voting had examined who is most likely to be a negative voter and why[18,19,21,23,24]. However, these questions are typically examined retrospectively (e.g., based on how voters had made their decision), which results in selection bias. Our studies also contribute to this literature by elaborating on the mechanisms underpinning such voting decisions, bridging political science research that has typically occurred at the population level with psychological research characterizing moment-to-moment decision dynamics within and across individuals. In doing so, we developed and validated a method that tailors voting decisions to an individual's

preferences, making it possible to study variability in voter behavior beyond the natural statistics afforded by real-world political choices.

On the surface, the idea of focusing voters on the candidate they want to reject brings to mind the well-known campaign strategy of focusing attention on a candidate's weaknesses (also referred to as negative campaigning[31–33]). Our work helps to formalize and differentiate potential influences of negative campaigning relative to other strategies like negative voting (i.e., rejection-based voting). Negative campaigning targets a change of relative desirability, seeking to influence voter choices by lowering the perceived desirability of the opposing candidate; yet, the effect of this can be to increase opt-outs rather than to increase support for the alternate candidate. This may help account for the mixed success of such strategies[31–33]. By contrast, rejection-based voting does not seek to alter a voter's preference but rather to express that preference by suppressing (or even reversing) the effect of overall desirability on voter opt-outs. Our simulations confirm this theorized difference between the two strategies (Supplementary Table 20; see Supplementary Text 1).

While our findings demonstrate that choice framing can alter decisions at the individual level when the choice itself is altered, it remains to be determined whether this extends to other methods for framing these decisions that do not involve changing the choice at the ballot box, such as messaging campaigns[34–36] aimed at reframing the choice in the individual's mind prior to the election. Separate from these potential implications for influencing voter turnout, our findings also have direct and more immediate implications for assessing public opinion. We show that a relatively simple alteration to how polls are framed has the potential to reveal the preferences of participants who may be masking those preferences because of their dislike of their options. While this may not produce a direct change in election outcomes, it can better inform the public about the true preferences of the electorate, which would have important downstream benefits for policy setting and even potentially for mobilizing voters who may otherwise misestimate preferences toward their preferred candidates or policies. Specifically, pollsters could consider adopting rejection-framed poll questions and explore their utility in estimating voter intentions/preferences and ultimately forecasting election outcomes (cf. Liu et al.[37]).

Our work also has implications for normative theories regarding negative voting[20,23] and broader theories of representations[38,39]. For instance, Kang[20] theorized that negative voting would favor a centrist candidate (i.e., less polarizing) over an extremist candidate (i.e., more polarizing). Our work brings empirical evidence and a mechanistic framework directly to bear on such predictions, including by simulating population-level election outcomes that could result from different distributions of voters with varying candidate preferences. While our work therefore speaks to the potential impact of negative voting on election outcomes, it is worth noting that negative voting could theoretically also have drawbacks not captured by our work, for instance facilitating so-called "ugly" (e.g., racially polarized) preferences[20], in the worst case producing a democracy with enhanced negativity and affective polarization[40]. The potential benefits of rejection-based voting suggested by our work should therefore be considered alongside any such potential costs.

Another noteworthy limitation of our work is that our samples were not explicitly designed to be representative of the US population in all respects, constraining our ability to generalize to this population (though see Supplementary Text 2 for robustness analyses controlling for available demographics). Rather, our studies enable stronger inferences about the general decision mechanisms that underpin decisions to vote (including the critical interaction between candidate desirability and choice frame), and about the role these mechanisms play in decisions to reveal one's preference when eligible voters are polled about actual political candidates. Nevertheless, it will be important for future work to test the extent to which our findings

generalize to all cross-sections of the American population, as well as to other countries, using approaches such as representative sampling and/or survey weighting[41].

Broadening out from their particular implications for voting decisions, our findings build on past work examining the effect of goal congruency on everyday choices, which showed that goal incongruency (selection among bad options or rejection among good options) leads to longer decision time and lower decision confidence[14–16,42,43]. These findings have been shown to reflect a basic property of decision-making—decisions about which option is best accumulate to a decision boundary (i.e., the point where a person feels ready to make a choice) most readily when choosing among good options; conversely, decisions about which option is worst accumulate to that decision boundary most readily when choosing between bad options[14–16]. Our work demonstrates that the influence of goal incongruency on deciding what to choose also generalizes to deciding not to choose[44–47]. By showing that unattractive options do not, in and of themselves, increase voter opt-outs, our work provides valuable insights into research on decision avoidance in broader contexts—from voting and polling for public opinion to deciding what to buy. Implications across psychology and political science are many, promising, and hard to reject.

## Methods
Here, we provide our experimental procedures, data analytic pipelines, and simulation settings to compare selection-based voting and rejection-based voting. We begin by describing the participant recruitment, inclusion, and exclusion, the two versions of the voting task, and the two preregistered poll surveys. We then outline our data processing strategies and statistical analyses. Finally, we describe how we simulate elections.

### Experimental procedures
The first three studies (two non-preregistered voting tasks and one preregistered survey study, named Studies 1, 2, and 3 in the following) were approved by Brown University's Institutional Review Board under protocol 1606001529. The fourth study (a preregistered survey, named Study 4) was approved by UC Berkeley's Committee for the Protection of Human Subjects under protocol 2024-06-17537. All participants were recruited on the Prolific platform (https://www.prolific.com/). The inclusion criteria for Studies 1 and 2 were those aged 18 to 55 and fluent in English. In Studies 3 and 4, in addition to the above two criteria, we used the pre-screening function (filtering by "U.S. Political Affiliation") provided by Prolific to recruit those who self-identified as Independents (see https://osf.io/djy4h for Study 3 and https://osf.io/89e75 for Study 4). We excluded participants who had participated in our previous decision-making-related studies from the recruitment. After signing the consent form, participants in Studies 1 and 2 were instructed to perform a voting task built up by the PsychoPy software (https://www.psychopy.org/) and completed a short Qualtrics survey afterward. In Studies 3 and 4, participants were directed to the poll survey after signing the consent form. After completing the survey, participants in all four studies were debriefed about the specific study's aim and directed back to Prolific for payment processes (8 US dollars for Studies 1 and 2; 1.5 US dollars for Studies 3 and 4). Consent was obtained for all participants in all four studies.

### Participants
Two samples of 100 participants were recruited on Prolific for Studies 1 and 2, respectively, in March 2024. We excluded data from those participants who either (1) decided to vote on all ballots without opting out once or (2) had equal to or more than 50% of ballots on which they opted to vote but chose the candidate inconsistent with their assigned goals (e.g., a participant assigned to Rejection condition but 60% of voted ballots they chose the best candidate would be excluded). 91 and

82 of the participants in Studies 1 and 2, respectively, were included for further data analysis.

In Study 1, 44 participants (Females/Males = 19/25, $M_{Age}$ = 37.0, $SD_{Age}$ = 9.31) were those assigned to the Selection condition ($N$ = 2 excluded by (1), additionally $N$ = 1 excluded by (2)), and 47 (Females/Males = 28/19, $M_{Age}$ = 37.3, $SD_{Age}$ = 9.79) were those assigned to the Rejection condition ($N$ = 5 excluded by (1), additionally $N$ = 1 excluded by (2)). In Study 2, 39 participants (Females/Males/Prefer not to respond = 14/23/2, $M_{Age}$ = 33.9, $SD_{Age}$ = 9.30) were those assigned to the Selection condition ($N$ = 5 excluded by (1), $N$ = 2 excluded by (2)), and 43 (Females/Males = 25/18, $M_{Age}$ = 33.6, $SD_{Age}$ = 9.84) were those assigned to the Rejection condition ($N$ = 7 excluded by (1), $N$ = 3 excluded by (2)). One participant who did not complete the study in Study 2 was also excluded. The assignment of conditions was based on a random seed automatically generated by the PsychoPy experimental code. Demographic information for Studies 1 and 2 is summarized in Supplementary Tables 1 and 2, respectively.

In Study 3, 1000 participants were recruited on Prolific on May 28th, 2024. Two participants completed the Qualtrics survey but failed to submit the completed Prolific study. The experimenter paid them manually, and their survey data was included. One participant submitted the Prolific study but lacked Qualtrics survey results. In the end, 1001 survey responses were collected. For the first prediction that we preregistered (see "Statistical Analysis"), we excluded data from those participants who did not report themselves as eligible voters (see "Survey Question"). 493 participants were those assigned to the Selection condition ($N$ = 6 excluded), and 485 were those assigned to the Rejection condition ($N$ = 17 excluded). For the second prediction that we preregistered, we further excluded data from those participants who reported "Prefer not to respond" in the poll question (see "Survey Question"). Additional 9 participants were excluded in the Selection condition (final $N$ = 484, Females/Males/Prefer not to respond = 251/230/3, $M_{Age}$ = 35.6, $SD_{Age}$ = 9.30); additional 2 participants were excluded in the Rejection condition (final $N$ = 483, Females/Males/Prefer not to respond = 237/242/4, $M_{Age}$ = 34.9, $SD_{Age}$ = 9.30). The assignment of conditions was based on the Qualtrics build-in function "Randomizer." Sample sizes of the two conditions were adaptively matched with the option "Evenly Present Elements" in Qualtrics being checked. Demographic information for Study 3 is summarized in Supplementary Table 3.

In Study 4, another 1000 participants were recruited on Prolific on September 6th, 2024. 1004 survey responses were collected, of which three came from participants who completed the Qualtrics survey but failed to submit the completed Prolific study. The experimenter paid them manually, and their survey data was included. One participant mistakenly did the survey twice in each condition. Two survey responses from this particular participant were discarded. In the end, 1002 survey responses from unique participants were included in the final analysis. After we excluded data from those ineligible voters, we had 494 who were assigned to the Selection condition ($N$ = 9 excluded) and 490 who were assigned to the Rejection condition ($N$ = 9 excluded). As we further excluded data from those "Prefer not to respond" responders, additional 5 participants were excluded in the Selection condition (final $N$ = 489, Females/Males/Prefer not to respond = 252/233/4, $M_{Age}$ = 34.6, $SD_{Age}$ = 9.38); additional 3 participants were excluded in the Rejection condition (final $N$ = 487, Females/Males/Prefer not to respond = 282/202/3, $M_{Age}$ = 35.5, $SD_{Age}$ = 10.2). The assignment of conditions was the same as in Study 3. Demographic information for Study 4 is summarized in Supplementary Table 4.

No statistical method was used to predetermine the sample size of Study 1, but our sample size per condition is larger than those in previous publications[14] ($Ns$' > 30). Confirming that our effect of interest (i.e., the interaction between overall desirability and choice goals on the probability of opt-out) was large, we adopted the same sample size in Study 2 as in Study 1 (see power analyses in "Statistical Analysis"). For

Studies 3 and 4, we based on a pilot study to calculate the sample sizes needed for our preregistered statistical tests with a power of 0.95 and a type-1 error rate of 0.05. The minimum sample size per condition is calculated as 220. Satisfying the power, we aimed to recruit 500 samples per condition (1000 samples in total per study). See the pre-registration for Study 3 (https://osf.io/djy4h) and Study 4 (https://osf.io/89e75) for details. Across four studies, the investigator who interacted with the study subjects was blinded to the subjects' assigned conditions.

## Voting task

In Studies 1 and 2, participants performed a voting task in which they could decide whether to vote between candidates or opt-out. Participants first saw 13 political issues (modified from Jenke and Huettel's work[48], see Supplementary Table 7 for a full list) and were asked two questions in a row for each issue: "Indicate your stance on (a specific issue, e.g., abortion) …" and "How important is this issue to you?" For each issue, participants first saw and answered the stance question; the importance question then came after the answer to the stance question had been submitted. Participants were asked to use sliding scales to answer both questions, with the scale ranging from −3 (left) to 3 (right). For the stance question, the descriptions on each end of the scale vary by issue: for example, abortion has its left end as "Legal" and right end as "Illegal;" spending on social security has its left end as "Decrease" and right end as "Increase." For the importance question, the descriptions are always "Not important at all" on the left and "Very important" on the right. See Supplementary Table 8 for descriptions and summary statistics of all stance and importance questions.

After participants had answered all the political issue questions, they went through instructions and practices before they entered the main task, in which they saw through a hundred ballots. On each ballot, participants in the Selection group were told to vote for the candidate they liked most, whereas those in the Rejection group were told to vote against the candidate they liked least. In both groups, participants could choose a "No Vote" option to opt out, such that they would opt out of the current ballot and move to the next one. In Study 1, participants would need to make a binary choice between vote (with "Choose" and "Reject" presented as options for Selection and Rejection groups, respectively) and no vote before they could select or reject one candidate; in Study 2, participants would make a trinary choice at once among two candidates and the No Vote option.

Each ballot featured two hypothetical candidates presented side by side, with two sliding scales and indicators revealing their positions on two political issues. In Study 1, participants used the arrow keys Up and Down to decide whether to vote; the options "Choose/Reject" and "No Vote" appeared on the top and bottom of the screen, with the locations randomized across participants. If participants opted to vote, they then used the arrow keys Left and Right to decide which candidate to select or reject. In the Selection group, the chosen candidate would be surrounded by a colored frame (yellow or green, consistent with the color of the candidate); in the Rejection group, the chosen candidate would be surrounded by a red frame and a red cross would be placed on the candidate figure, indicative of being rejected. To prevent online participants from spending infinite time on the task, we set an implicit 2-min deadline for each trial (ballot) without informing participants beforehand. The trial would automatically end and move to the next ballot if the deadline was hit, which were rare events in both Studies 1 and 2 (Study 1: 0.10%, Study 2: 0.16%).

To generate ballots for each participant, we first ranked 13 issues based on each participant's importance ratings and used the 1st to 8th issues for ballots in the main task, the 9th to 12th issues for the ballots in the practice session, and discarded the issue which a participant

regarded as least important. Out of the top 8 important issues, we created a table of 252 hypothetical candidates with possible combinations of two issues ($8 \times 7/2 = 28$ unique pairs) and stances: each issue could have three possible levels (left, neutral, right) of stances with random noise. Out of the 252 candidates, we generated 100 candidate pairs presented on the ballots for each participant with the following algorithm: first, we randomly sampled from the space of overall and relative desirability of candidates (cf. Supplementary Fig. 1b) constrained by each participant's own ratings and derived the corresponding desirability of the two candidates. We then looked up the table of 252 candidates to find the two candidates that had the closest desirability (for individual candidate desirability, see "Desirability of Candidates") under the constraint that the two candidates expressed views on four unique issues. We repeated the above procedure once while ensuring that the four issues used did not overlap with those in the previous candidate pair. Consequently, we had two ballots with four candidates, and 8 issues were all being used. Through 50 iterations, we generated 100 candidate pairs that controlled the occurrence of issues, with every issue occurring exactly 50 times. The order of 100 candidate pairs that participants would see on the ballots was randomized. As a result, we also set up distributions of overall and relative desirability of candidates that were similar across participants and conditions (Supplementary Fig. 1b). In addition, we counterbalanced the candidate pairs such that the desirability of the candidate on the left was higher than that on the right in half of the ballots. We used yellow and green colors to make the two candidates in a ballot distinct from each other, and the colors of the candidates were counterbalanced across the task.

In the practice session, participants went through an example of 8 ballots, with the arrangement of issues on the ballots pre-defined in an Excel table. To familiarize participants with all the possible responses and corresponding keys, we instructed them on whether to vote or opt-out and which key should be pressed for each ballot. After the practice session, we set up a quiz to remind participants of their assigned goals. A sample question was, "Imagine you prefer the green candidate over the yellow candidate; which one should you vote for (against)?" For the above question, participants in the Selection group should choose the green candidate, while participants in the Rejection group should choose the yellow candidate. Participants needed to make 4 correct responses in a row (or hit an upper limit of 20 quiz questions being asked) to pass the quiz and move forward to the final reminders preceding the main task.

## Desirability of candidates

With participants' stance and importance ratings, we constructed their desirability for each candidate as follows[48]:

$$D_{i,j} = \frac{10}{12} \sum_{k=1}^{2} \text{importance}_{i,k} * \left[ 1 - \frac{\text{abs}\left(\text{stance}_{j,k} - \text{stance}_{i,k}\right)}{6} \right] \quad (1)$$

Where $D_{i,j}$ was the desirability a participant $i$ would hold for a candidate $j$, which was a weighted sum of the alignments of participants' positions (stance$_{i,k}$, from −3 to 3) with candidates' positions (stance$_{j,k}$, from −3 to 3) on the two presented issues ($k = 1, 2$). The smaller the absolute difference between the two, the larger the alignments yielded and the higher the desirability a participant had toward a candidate. The weights reflected how important (rescaled to the range of 0–6) the issues were to that participant. Appropriate scaling factors were applied so that the desirability eventually ranged from 0 to 10.

## Survey questions

In Studies 1 and 2, we asked participants about their party affiliations and demographics after they completed the voting task. For the party affiliation question, we adopted from Jenke and Huettel's work

(2020)[48], "Generally speaking, in politics do you consider yourself as:" Response options ranged from "−3: Strong Democrat", "0: Independent," to "3: Strong Republican." For the demographic questions, we asked (1) Sex (referred to as the sex assigned at birth), (2) Age, (3) Race, (4) Race: Hispanic or Latinx, (5) Education years, (6) their biological mother's education level, and (7) their biological father's education level.

In Studies 3 and 4, we asked participants a poll question about their voting intentions toward the 2024 US presidential candidates. Participants assigned to the Selection (Rejection) condition in Study 3 were asked: "At this time point, who would you be most inclined to vote for (against) in the upcoming US general elections?" They were presented with four options: (a) I would vote for (against) Joe Biden, (b) I would vote for (against) Donald Trump, (c) I'm undecided, (d) Prefer not to respond. In Study 4, participants assigned to the Selection (Rejection) condition were asked: "In a two-person race, who would you be most inclined to vote for (against) in the upcoming US general election?" They were presented with four options: (a) I would vote for (against) Kamala Harris, (b) I would vote for (against) Donald Trump, (c) I'm undecided, (d) Prefer not to respond.

After the poll question, participants in both studies were asked about their eligibility to vote: "Are you eligible to vote in the upcoming US general elections?" They were presented with four options: (a) Yes, (b) No, (c) Not sure, (d) Prefer not to respond. Participants were then asked about their (1) Likelihood of voting (from −3 to 3) in the upcoming election and (2) Favorability (from 0 to 100) toward the two candidates. In Study 3, participants were further asked about their (3) Trust toward Joe Biden or Donald Trump (from −3 to 3) on six policy issues: The Economy, Foreign Policy, Immigration Policy, Health Insurance, Gun Buying, and Abortion. See Supplementary Tables 5 and 6 for descriptive statistics of each question. Finally, we included questions about participants' party affiliations and demographics, which were the same as in Studies 1 and 2 (A shortened version was used in Study 4; See Supplementary Tables 3 and 4). See the preregistrations (Study 3: https://osf.io/djy4h; Study 4: https://osf.io/89e75) for all the wording of questions and options.

## Statistical analysis
**Data preprocessing.** In Studies 1 and 2, we centered and scaled each ballot's overall desirability, relative desirability, and trial order (1st to the 100th ballot) with respect to each participant before feeding them to the regression models. A ballot's overall desirability was defined as the mean desirability of the two candidates; the relative desirability was the difference between the desirability of two candidates (signed if denoted as "$X$ vs. $Y$"; unsigned otherwise).

When analyzing candidate choices (i.e., when participants opted to vote), we excluded both ballots on which participants decided to opt out of voting and ballots on which participants spent too long on the ballot and hit an implicit deadline (2 min; see "Voting Task"). When analyzing opt-outs, we regarded those ballots on which participants spent too long and hit the deadline as not abstaining. For both Studies 1 and 2, decision time (choice RT) was defined as the time participants spent after seeing the two candidates on the screen until they chose one candidate by pressing the Left or Right button. We log-transformed (with 10 as the base) choice RTs to reduce skew and improve normality before feeding them to the regression models.

**Statistical models.** The results of all regression models are summarized in Supplementary Tables 9–17.

For Study 1, we used generalized linear mixed-effect regression (R package lme4[49]) to analyze candidate choices for those ballots when participants opted to vote. For both Selection and Rejection conditions, we ran regressions using the dependent variable of whether the left candidate was chosen (chosen: 1; unchosen: 0) and included the relative desirability (the desirability of the left candidate minus that of

the right candidate), overall desirability, trial order, and the interaction between the trial order and the relative/overall desirability, with random (participant-specific) intercept and slopes for each predictor. We also ran linear mixed-effect regression to analyze choice RT (log-transformed) using the same predictors above and the random (participant-specific) intercept and slopes of all the predictors. Finally, we ran generalized linear mixed-effect models to analyze opt-out decisions (opt-out: 1; not opt-out: 0) using the same predictors and random structures. All the reported models converged well (relative maximum gradient <0.001).

We ran three additional regression models (candidate choices, choice RTs, and opt-out decisions) for each study, combining participants in both conditions to report the interaction effect (Supplementary Tables 13, 14, and 17). Those regressions included the choice condition (Selection: −1, Rejection: 1), overall desirability, relative desirability, trial order, and the interaction between the choice condition and overall/relative desirability, the trial order and overall/relative desirability, and the three-way interactions (choice condition X trial order X overall/relative desirability) with random (participant-specific) intercept and slopes for each predictor. All the reported models converged well (relative maximum gradient <0.003).

For Study 2, we used the same setup of regression models and predictors as in Study 1. All the reported models converged well (relative maximum gradient <0.001), except for the model of candidate choices in the Rejection group (relative maximum gradient = 0.066). To get a better convergence of the model, we reduced the full random structure by removing the random effect of the interaction between relative desirability and trial order (relative maximum gradient <0.001). The estimated main effects of relative desirability and trial order had negligible changes from the full random-structure model.

To examine the robustness of our findings in Studies 1 and 2, we ran power analyses on our core effect of interest, that is, the significant interaction between the choice goals and overall desirability on the probability of opt-out. Using simulation-based power analysis (R package: simr[50]), we find that both studies require a sample size of $N > 45$ to achieve a power of at least 0.8; both studies had Ns of at least 82, with projected power saturating to 1 (CI = [0.996, 1] for both studies). To be conservative, we set the estimate of actual effect size to be 0.5, way below the observed values (Study 1: log odds = 1.71; Study 2: log odds = 1.54; Supplementary Table 17), and find that both studies still achieved a power of at least 0.86 (Study 1: power = 0.86, CI = [0.837, 0.881]; Study 2: power = 0.936, CI = [0.919, 0.950]). In addition, we calculated the sign error rates and magnitude errors[51] using the same conservative estimate, and confirmed that our study was immune to both kinds of error (sign error rates <$10^{-6}$, magnitude errors <1.1, comparable to typical public health studies[51]).

For Study 3, our primary, pre-registered prediction was that eligible voters in the Rejection condition were more likely to vote (less likely to opt-out) than those in the Selection condition. To test this, we coded participants who responded "I'm undecided" or "Prefer not to respond" as uncommitted voters and conducted a two-proportion, one-tailed $z$ test to compare the proportions of uncommitted voters in two conditions. We used the standard $p < 0.05$ for this test. To reduce noise, we further excluded uncommitted voters who chose "Prefer not to respond" in both conditions and conducted another two-proportional one-tailed $z$ test. We used stricter criteria, alpha = 0.025, for this test. The $p$ value for the second test was reported in the main text, and the detailed results for the two tests are included in Supplementary Table 18.

For Study 4, we preregistered the same two predictions as in Study 3. The $p$ value for the second test was reported in the main text, and the detailed results for the two results are included in Supplementary Table 19.

**Statistical assumptions.** Group differences in continuous measures between the Selection and Rejection conditions (e.g., Age; Supplementary Tables 1–6) were tested using the Brunner–Munzel test[52], a non-parametric two-sample test that does not require normality and equal-variance assumptions. To test group differences in count measures (e.g., Sex; Supplementary Tables 1–4), we used Fisher's exact test[53]. No adjustment of multiple comparisons is applied in this work.

**Software.** All analyses and simulations (see below) were performed in R (version 4.4.0). Mixed-effect model fitting was performed using the R packages lme4[49] (version 1.1.37) and lmerTest[54] (version 3.1.3; specifically used to calculate statistics, $p$ values, and degrees of freedom in linear mixed-effect models). Power analyses for mixed-effect models were performed using the R package simr[50] (version 1.0.7). Brunner–Munzel tests were performed using the R package brunnermunzel[55] (version 2.0), and Fisher's exact tests were performed using the R core package stat.

## Simulations
**A voting agent.** We used the regression models of candidate choices and opt-out decisions fitted by our choice data in Studies 1 and 2 (see "Statistical Analysis"; Fig. 4 and Supplementary Fig. 5) to simulate agents that could either vote according to different goals or opt out. We used the models that only included the fixed effects to simulate a voting agent without idiosyncratic behaviors biased toward any individual participant. We simulated candidate choices (which candidate was chosen) and opt-outs (whether the vote was cast) in response to 81 distinct candidate pairs (9 × 9, desirability from 1 to 9 out of 10 points for each candidate) in both Selection and Rejection conditions. The same preprocessing (centering and scaling) was performed, with the mean and variance of predictors (overall and relative desirability) averaged across all participants as centers and scaling factors, respectively. The trial order was set as 1, simulating that the voting agent only has one ballot to vote on. The model outputs were the probabilities of choosing each candidate when votes were cast and the probabilities of opt-out.

**Elections.** With our voting agent models, we simulated projected electoral outcomes by setting up two populations of voters with their preferences for the two candidates distinct from each other. The Blue Voter preferred the Blue Candidate over the Red Candidate (with the corresponding candidate desirability 4 versus 1), while the Red Voter preferred the Red Candidate over the Blue Candidate (with the desirability 6 versus 3). We simulated four elections, with each having 10000 voters in total, and set up four proportions of Blue versus Red Voters: (Blue, Red) = (500, 9500), (3500, 6500), (6500, 3500), (9500, 500). The projected electoral outcomes were the difference between the actual votes for the two candidates. For selection-based voting, the actual votes for the Blue Candidate would be the product of (1) the probability of not opting out, (2) the probability of selecting the Blue Candidate, and (3) the number of votes from both the Blue and Red Voters; for rejection-based voting, the votes to reject the Blue Candidate would be counted as the votes for the Red Candidate, calculated as a product of (1) the probability of not opting out, (2) the probability of rejecting the Blue Candidate, and (3) the number of votes from both the Blue and Red Voters.

## Reporting summary
Further information on research design is available in the Nature Portfolio Reporting Summary linked to this article.

## Data availability
The de-identified experimental and survey data generated in this work are publicly available at https://doi.org/10.5281/zenodo.17993607[56].

## Code availability
Data analysis and simulation scripts used in this work are publicly available at https://doi.org/10.5281/zenodo.17993607[56].

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

## Acknowledgements

The work was funded by a grant from the National Science Foundation (Collaborative Research in Computational Neuroscience Award #2309022) to A.S. The authors are grateful to Libby Jenke for providing materials on policy issue questions, to Maximilien Boucher, Daantje de Bruin, Liz Liyu Chen, Meriel Doyle, Mahalia Prater Fahey, and Eva Swartz for assistance and advice pertaining to experimental design, and to Molly J. Crockett, Oriel FeldmanHall, Uma R. Karmarkar, Xiamin Leng, David G. Rand, and Dmitry Taubinsky for helpful feedback.

## Author contributions

A.S. and Y.-H.S. conceived the study. Y.-H.S. programmed the tasks and surveys, collected and analyzed the data. A.S. and Y.-H.S. interpreted the results, wrote the manuscript, and edited the manuscript.

## Competing interests

The authors declare no competing interests.
