## [Transparent Peer Review file · Nature Communications]

Rejection-based choices discourage people from opting out of voting

Corresponding Author: Professor Amitai Shenhav

Version 0:

Reviewer comments:

Reviewer #1

(Remarks to the Author)

This interesting and well-developed series of experimental studies sheds new light on turnout in democratic elections. Grounded in “negative voting” theories, the authors’ main intuition is that voters will be less likely to opt-out from the electoral process if they were asked to explicitly vote “against” rather than “for” in either fictional or real (e.g., US 2024 Presidential) elections. The various experiments convey converging results. Indeed, turnout appear to increase when voters are asked to pick a candidate to vote against. My reading of this paper is generally favorable, though I would like to raise the following points:

1. The theoretical underpinnings of the paper are thin, at best. The authors relate to turnout literature and to its indifference vs. alienation approaches, to then move straight into positive vs. negative voting. Considering the very wide implications of their findings (see below), the authors should in my view make a better effort in framing their arguments within broader theories of representation. What type of democracy underlies elections in which people reject rather than select?
2. The sample size of studies 1 and 2 is sufficiently powered for across-subject (select vs. reject groups) comparisons. However, it does not provide enough leverage to illuminate on the individual characteristics of the respective opt-out populations. What type of voters are left out in a selection vs rejection type of election, respectively? I agree that the proportion of opt-out diminishes in the latter type, but it would be crucial to understand if the respective opt-out groups differ among themselves, and if so, on which key socio-political characteristics.
3. Study 3 is possibly the most ingeniously deployed of the four. However, I found it harder to follow its numerous steps through the three brief paragraphs provided by the authors. Figure 4’s caption comes in handy in this respect, but I wonder if the authors could offer a better, and more integrated, narrative throughout this subsection.
4. As rightly argued by the authors, this study’s implications “across psychology and political science are many, promising, and hard to reject”. However, I found very little discussion of this crucial point. Especially when it comes to political science, I would like a deeper reflection of the relative pros and cons of rejection-based electoral processes. Normative theories of democracy are widely based on selection, rather than rejection, when it comes to the relationship between voters and elected in a representative democracy. Hence, an alteration of the basic premise of voting would carry enormous implications. Negative campaigning (only briefly touched in the discussion) could actually increase and becomes the normality of campaigning (also considering that out-party hate has turned stronger than in-party love across established democracies worldwide). This could in turn affect patterns of affective polarization in the electorate, with possible spill-over effects in terms of political intolerance and even acceptance of political violence. In my view, the authors should take these concerns seriously. As a matter of fact, turnout may decline in rejection-based elections. But at what cost for the democratic process? In what ways would this change affect the socio-political composition of abstainers?

Reviewer #2

(Remarks to the Author)

Across two experiments the authors test whether opt out rates in a voting choice task are driven more by indifference, not wanting to vote for either of two bad choices, or by the mismatch between the goal of wanting to choose the best candidate

versus the task of choosing between the lesser of two unappealing options (when no good candidate is on offer). The authors manipulated both the absolute undesirability of each of two candidates, as well as their relative desirability (operationalized as issue alignment with the participant), across a series of 100 choices. The authors also manipulated whether participants were asked to select the preferred or reject the dispreferred candidate (between subjects). The critical outcome variable was rates of opting out of voting across the 100 choices.

WHEN they voted in Exp 1, participants selected the preferred and rejected the dispreferred candidate about ¾ of the time on average. These choices increased/got faster as the desirability gap between the candidates grew and made it easier to distinguish them.

However, participants opted out 40% of the time in the select condition. This was more likely when the candidates were similar, and more importantly, much more likely when the absolute desirability was low for each candidate.

Opting out rates were lower in the reject condition: 33%. Critically, they were no more likely to opt out in the low-low ballot cases. If anything they were more likely to opt out in high-high choices but those rates never approached the rates in low-low/select.

Exp 2 replicated 1 despite the task change which put the opt out choice at the same decision stage as selecting/rejecting a candidate.

A simulation building on Exp 2 points to important consequences of these outcomes: opt-out votes are less correlated with absolute candidate desirability. This matters because it means opt-out instances won't cluster in one region (or party, in this case) along the candidate desirability gradient (Fig 4).

Two follow up surveys with independents converged with the basic finding: fewer opt outs in rejection than selection frames.

It's a great paper. It's beautifully written, the experiments and surveys are simple and well designed, and the effect sizes are "huge" (as the authors note). If I was just reading this paper without a specific journal in mind, I'd say minor revisions (see below), if that.

Two things arise, however, when thinking about this paper in the context of Nature Comm. With respect to the work's theoretical implications in political science, I'm having trouble figuring out what readers have learned here that hasn't already been discussed in the negative voting and ranked-choice voting literatures (including the Pew data the authors cite here: [24]). With respect to the work's theoretical implications in decision-making, I'm having trouble convincing myself that these findings constitute more than an application of this research group's previous elegant work on goal congruency in value. As for practical applications: in a vacuum the work has implications for how we run elections. But the reality is that changing voting laws and procedures in the U.S. (the survey case here) is incredibly difficult. Unless the authors are thinking of testing these ideas in other democracies, it's hard to see what the practical implications of this work are, just given realistic constraints.

More specific questions:

Why restrict the study participants to independents who make up the smallest group of voters in the electorate relative to the two parties? Unless I missed something, the authors didn't apply that same restriction to the experiment samples.

Related, how was independent defined? Only those who answered 0 on the Jenke & Huettel scale?

Reviewer #3

(Remarks to the Author)

This is an interesting paper which evaluates the extent to which survey respondents' willingness to express ranked preferences over political candidates varies as a function of whether they are asked to make positive selections (e.g., which of these candidates is better) or negative selections (e.g., which of these candidates is worse). Fielding an interesting experimental design to two small samples of respondents, the authors demonstrate that rejection-based choice questions result in lower levels of abstention in hypothetical voting decisions than questions based on acceptance-based wording. They also demonstrate that the degree to which these types of question lead to opt out vary in predictable ways with a priori measurements of candidate quality (such that opt out rates are higher under acceptance-based selection when both candidates are low quality, and opt out rates are higher under rejection-based selection when both candidates are high quality). In a second set of experiments, they then demonstrate that these dynamics hold in a real-world setting for independent voters considering voting decisions in the context of the US presidential election.

The paper is well-written and highlights an interesting pattern of responses in the context of political surveys. However, I have some concerns about the study design and interpretation which I think need to be addressed if the authors are offered the opportunity to resubmit the manuscript.

1. My primary concern with this paper is that the first two experiments rely on very small and unrepresentative samples. Study 1 has N = 91, while Study 2 has N = 100, both conducted via Prolific. The inferential dangers of using small samples of this nature are well known, not just in terms of the potential for imbalance on unobserved covariates but also in their propensity to produce both type S and type M errors (e.g., <https://journals.sagepub.com/doi/10.1177/1745691614551642>).

Related to this point, I am concerned about the power of the design. I wanted to know whether the authors had pre-registered this design (my impression is that they pre-registered the central expectations for experiments 3 and 4, but not the expectations or the analysis strategy for experiments 1 and 2). If they did pre-register the experiment, did they conduct a power analysis for these designs? The effects they report are very large but given that small sample studies are more like to suffer from both sign and magnitude bias, I think it is important to establish the extent to which the authors had been anticipating effects of this magnitude in advance of running the study.

In addition, the descriptive statistics in the appendix suggest that treatment and control groups in both of the first two studies are not well-balanced in terms of key demographic variables such as gender and parental education. This provokes the concern that there may also be other, unobserved differences between treatment and control groups that are driving differences in observed outcomes, rather than the rejection/acceptance prompt condition itself. Ultimately, my impression is that very small online surveys of the sort reported here are generally not sufficiently robust to warrant publication in Nature journals because of the inferential problems associated with them, but I am open to being convinced otherwise. Given these limitations, I think that -- at the very least -- the authors need to explicitly acknowledge the risks posed by small, imbalanced samples and discuss how these limitations might affect the interpretation of their results.

2. On representativeness: Unless I have misunderstood, I do not believe that these samples are collected such that they are representative of the broader US population and nor do the authors appear to use survey weighting approaches in any of their analyses. These issues therefore raise concerns about the subsequent generalizability of the findings presented by the authors to the ultimate population(s) of interest. Again, I think this needs to be explicitly addressed in the manuscript.

3. I didn't understand the authors' decision to include only independent voters in the third and fourth study. By focusing attention on this subset of the American electorate, the authors unnecessarily restrict the external validity of their study. In addition, I am concerned that it is for this group of voters for whom we would expect to see the biggest differences between the acceptance-based and rejection-based choice conditions (because committed Democrats and Republicans are more likely to hold fixed attitudes over the Trump/Biden candidacies). In essence, then, the differences reported in figure 5 are therefore likely to be upper bounds on the size of the effects the authors are interested in. I think this needs more explanation and clarification in the paper.

4. I also think the authors need to do more to explain the normative importance of distinguishing between the "indifference" and "alienation" explanations for non-participation which is central to their framing of the paper. They suggest that -- when alienation is the motivating factor behind non-response -- this is important because "the failure to reveal their preferences can be highly consequential". But in a voting context, in which voters are deciding between two options that they dislike, it is not clear that it *is* consequential, as in either case we lack the information from voters about their relative preferences over candidates. I wondered therefore whether the authors have something more particular in mind here, which would be worth describing in more detail in the paper.

5. In general I found the simulation analysis presented in the paper underwhelming. The authors use this analysis to show that reducing levels of abstention by asking rejection-based selection questions can produce election outcomes that are more representative of average voter preferences. However, this seems almost trivially true, as it follows directly from the fact that there are substantial differences in abstention across the experimental conditions. There is an enormous literature on the effects of non-random turnout variation on election outcomes, and the simple simulation that the authors employ doesn't really add any new intuition or insight, I didn't think. What do the authors think they are contributing through this analysis that is not obvious from the original presentation of their experiments? If the answer to that question is "not very much", then I'd suggest dropping the simulation analysis from the paper.

6. It would be helpful for the authors to link their findings to the literatures on political polling and election forecasting. For instance, one of the interesting areas of application for their rejection-based question form would be in prospective polling of vote intentions. Does asking these types of questions help to improve election forecasting accuracy? If so, that would be a practically useful contribution of the paper that is not currently explored.

Version 1:

Reviewer comments:

Reviewer #1

(Remarks to the Author)

The revised version of the manuscript has effectively addressed all the concerns raised in my review.

Reviewer #2

(Remarks to the Author)

As I noted last round, I really like this paper. I still do. I also appreciate the authors' thoughtful replies to all three reviews. The revision addresses all but one of my concerns: specifically related to the generalizability of the results.

Yes, random assignment to condition with a large enough sample will address a lot of concerns about causal inference. We generally reserve concerns about the representativeness of the sample for challenges to external validity. However, based on how small the samples are in these studies, plus some recent experiences in our own research (where we have

observed completely different results in politics-related experimental contrasts, holding N constant, but depending on whether we've run the study with a convenience versus representative sample), I want to second R3's first concern.

The results aren't only limited to independents, they are limited to a likely highly biased subset of them. It isn't an accident or arbitrary norm in political science that even the experiments are run with representative samples these days. To the extent that the authors want to claim that their work has implications for campaign framing and election forecasting, the evidentiary threshold is quite a bit higher.

Reviewer #3

(Remarks to the Author)

I appreciate the authors engaging constructively with my comments and for implementing the changes they have made in response to both my own review and that of the other reviewers. I'm happy to recommend the article is accepted on the basis of their revisions.

I've made some comments on each of the relevant revisions below, in case it is of use to the authors.

1. Power: I am convinced by the authors' response in the memo about the power of the design of their initial experiments, at least with respect to detecting the relatively large effect sizes that they focus on. While using the estimated effect sizes from the two studies to calculate power retrospectively (as the authors do in lines 444-455 of the revision memo) is not my preferred option, as the effect size from single, low-N studies is noisy and can lead to dramatic overestimates of power (see, for example, <https://pubmed.ncbi.nlm.nih.gov/29994928/>), I am more convinced by the subsequent analyses which uses the findings from additional studies on decision avoidance to inform the power analyses.

2. Pre-registration: The authors did not directly address my question about pre-registration, which does prompt me to repeat it. Were studies 1 and 2, and the associated analysis strategy pre-registered? Given the findings on power above, I do not think this is critical but it would be worth making sure it is clear in the paper (I find the description on page 15 a little vague here).

3. On imbalance: I appreciate the authors including the additional analyses in supplementary text B and am convinced by this analysis.

4. Representativeness: The text on representativeness on page 14 is helpful, thanks.

5. Including only independents: This clarification is also helpful.

6. Alienation vs indifference: The new text is really useful for illustrating the normative importance of the two accounts.

7. Simulation analysis: Fair enough. I still don't find that this analysis too much on top of the other empirical analyses but it is the authors' paper, not mine!

8. Election forecasting/polling: I think this paragraph is a helpful addition.

Replies to Reviewer 1:

This interesting and well-developed series of experimental studies sheds new light on turnout in
democratic elections. Grounded in “negative voting” theories, the authors’ main intuition is that
voters will be less likely to opt-out from the electoral process if they were asked to explicitly vote
“against” rather than “for” in either fictional or real (e.g., US 2024 Presidential) elections. The
various experiments convey converging results. Indeed, turnout appear to increase when voters
are asked to pick a candidate to vote against. My reading of this paper is generally favorable,
though I would like to raise the following points:

1. The theoretical underpinnings of the paper are thin, at best. The authors relate to turnout
literature and to its indifference vs. alienation approaches, to then move straight into positive vs.
negative voting. Considering the very wide implications of their findings (see below), the authors
should in my view make a better effort in framing their arguments within broader theories of
representation. What type of democracy underlies elections in which people reject rather than
select?

We apologize for not clarifying the foundation of our study sufficiently. While we motivated our
research with its link to population-level findings in political science literature (e.g., turnout;
alienation vs. indifference; positive vs. negative voting), we see our work as advancing research
on at least two fronts:

A) First, our work builds on recent empirical and theoretical findings in decision science, which
show that participants weigh information about their options (e.g., consumer goods) differently
depending on their choice goal¹⁻⁴. This research has yet to examine the dynamics of voting
decision, nor to generate and test predictions about decision avoidance such as opting out (in
any choice domain). Our work bridges both of these gaps. We summarize our foundation and add
the paragraphs below to the updated Introduction section:

We built our work on recent theoretical and experimental findings in decision science,
which show that participants weigh information about their options (e.g., consumer
goods) differently depending on their choice goal¹⁻⁴. When facing multiple options they
like, participants choose faster when their goal is to select the best option than the worst
one. When facing multiple options they dislike, they are faster to choose the worst one
than the best one. These findings have been accounted for by models that characterize
decision-making as a process of evidence accumulation towards a decision threshold,
whereby evidence congruent with one’s decision goal (choose worst vs. best) reaches
threshold fastest³⁻⁵.

Our current work builds on these previous findings in two important ways. First, by testing
whether goal-driven reversals in choice dynamics generalize from consumer choice to

voting decisions (e.g., resulting in faster choices when selecting the better of two good
candidates or choosing the worse of two bad candidates). Second, by testing, for the first
time, whether these distinct choice goals also invert patterns of decision avoidance (i.e.,
whether to choose or opt-out) depending on the quality of one's options. If decisions
about whether to choose are driven by goal congruency in the same way as decisions
about which option to choose, then decision-makers would be expected to opt-out more
or less depending on whether the quality of their option set aligns with their choice goal.
In the context of voting decisions, this would predict participants opting not to vote when
being asked to choose the better of two candidates they dislike, but not when asked to
reject the worse of the two.

B) The second way our work advances past research is by grounding normative theories regarding
negative voting^{6,7} and broader theories of representation^{8,9} in cognitive and psychological
mechanisms. We agree with the reviewer that we could have unpacked this connection in our
original manuscript, and have now revised the manuscript with this in mind (see below).

Briefly, we now note, for example, that our simulated elections are consistent with a key
prediction by Kang's (2010)⁶ normative account of negative voting that a centrist candidate (i.e.,
less polarizing) would be favored over an extremist candidate (i.e., more polarizing) under
rejection-based voting. Therefore, imposing rejection-based voting may lead to fewer extremist
candidates being elected and, hence, a less polarized democracy^{10,11}. However, as the reviewer
also pointed out, it is equally valid to predict that imposing rejection-based voting may lead to
more negative campaigning and/or what Kang (2010) called "ugly" (e.g., racially polarized)
preferences, hence, a democracy with enhanced negativity and affective polarization¹². The
potential benefits of rejection-based voting suggested by our work should therefore be
considered alongside any such potential costs. We have included the above points in the updated
Discussion section:

Our work also has implications for normative theories regarding negative voting^{6,7} and
broader theories of representations^{8,9}. For instance, Kang (2010)⁶ theorized that negative
voting would favor a centrist candidate (i.e., less polarizing) over an extremist candidate
(i.e., more polarizing). Our work brings empirical evidence and a mechanistic framework
directly to bear on such predictions, including by simulating population-level election
outcomes that could result from different distributions of voters with varying candidate
preferences. While our work therefore speaks to the potential impact of negative voting
on election outcomes, it is worth noting that negative voting could theoretically also have
drawbacks not captured by our work, for instance facilitating so-called "ugly" (e.g., racially
polarized) preferences⁶, in the worst case producing a democracy with enhanced

negativity and affective polarization¹². The potential benefits of rejection-based voting
suggested by our work should therefore be considered alongside any such potential costs.

2. The sample size of studies 1 and 2 is sufficiently powered for across-subject (select vs. reject
groups) comparisons. However, it does not provide enough leverage to illuminate on the
individual characteristics of the respective opt-out populations. What type of voters are left out
in a selection vs rejection type of election, respectively? I agree that the proportion of opt-out
diminishes in the latter type, but it would be crucial to understand if the respective opt-out
groups differ among themselves, and if so, on which key socio-political characteristics.

The reviewer raises an excellent question about sources of individual variability in opt-out
behavior, and one which we think is worthy of further exploration. However, it is also beyond the
scope of the current work. As we now clarify above, our goal was to establish a general
mechanism for decisions to opt out of voting, and to demonstrate that laboratory and real-world
opt-out behavior changes drastically with choice framing. Our findings provide strong evidence
in support of both aims. Having said that, while our studies were not designed to further examine
individual variability in opt-out behavior (which, as the reviewer notes, would typically require
much larger samples), two aspects of our work are noteworthy for future research aimed at
investigating such variability.

First, though we are limited in providing positive evidence for sources of individual differences in
our observed behavior, we have performed comprehensive analyses aimed at testing whether
our findings of framing effects are robust to controlling for a wide range of individual-level
variables collected across our studies, including measures of demographics (e.g., sex, race) and
political affiliation. These analyses are now reported in Supplementary Text B and Extended Data
Table 23.

Second, a key contribution of our work is the development and validation of the first laboratory
task that measures variability in within-individual voter opt-out decision-making across
individually-tailored levels of overall and relative candidate desirability. This task enables future
researchers to examine variability across individuals, not only at the level of average opt-out
decisions but also in the sensitivity of both opt-out and candidate choices to overall and relative
candidate desirability. Such analyses, which are included in the supplement noted above, can
provide a richer understanding of how people differed in their choices as well as in the
component processes by which these decisions were formed, paralleling research on individual
differences in other choice domains^{13,14}.

3. Study 3 is possibly the most ingeniously deployed of the four. However, I found it harder to
follow its numerous steps through the three brief paragraphs provided by the authors. Figure 4's
caption comes in handy in this respect, but I wonder if the authors could offer a better, and more
integrated, narrative throughout this subsection.

We apologize that this section was previously hard to follow. We have now revised and expanded
the section for greater clarity, including by rearranging descriptions and providing more signposts
to different components in Figure 4:

When simulating conditions in which all agents are forced to select which of the
candidates they prefer (i.e., no possibility of opt-outs, or "full turnout"; Figure 4a, left
panel), we see that the likelihood of the Blue Candidate receiving a given agent's vote
(the blue-red gradient) depends primarily on how much better that candidate is than the
other candidate (relative desirability; deviation from the diagonal line), independent of
how much the voter likes the Blue Candidate (overall desirability; deviation from the
origin). If we simulate the same elections/polls in a world in which voter opt-outs are
permissible (Figure 4a, middle panel), we see that such opt-outs would occur
disproportionately in cases where agents find both candidates unappealing (alienated
voters; white areas in the bottom-left corner), even if one of these candidates is
consistently less unappealing than the other.

These findings imply that, as a result of their increased opt-out rates, voters who dislike
both candidates will have less influence on the ultimate preference measures (Figure 4a,
right panel). This can be demonstrated by simulating two hypothetical sets of voters who
have opposite rankings of the two candidates (Figure 4c): Blue Voters, who prefer the
Blue Candidate over the Red Candidate by a 3-point margin; and Red Voters, who prefer
the Red over the Blue Candidate by an equivalent (3-point) margin. The only difference
between these two voter groups is that the Blue Voters like both candidates less (rating
the Red Candidate as a 1 and the Blue Candidate as a 4), whereas the Red Voters like both
candidates more (6 for Red vs. 3 for Blue). In a case like this, we find that even when the
majority of the population are Blue Voters, because these voters are more prone to opt-
out, the election/poll can result in a win by the Red Candidate (Figure 4d, lighter bars).
The same bias would hold in reverse if the majority of the population were Red voters
who preferred both candidates less than Blue voters; in either case, the voters who view
the candidates as more desirable would be predicted to benefit from an advantage in
representation.

These cases of preference measures deviating from population preferences occur when
simulating traditional, selection-based choice (i.e., choosing the better candidate). When
we instead simulate voting behavior under rejection-based choice (Figure 4b), we find

that preference measures are much more faithful to the preferences of the population of
eligible voters (Figure 4b, right panel), such that the Blue versus Red Candidate is
expected to win when Blue versus Red Voters are in the majority, largely irrespective of
the overall desirability of the candidates (Figure 4d, darker bars). This results from the
diminished impact of overall desirability on opt-outs by individual voters and the
increased turnout of alienated voters (Figure 4b, middle panel). Consequently, an agent's
likelihood of voting for the Blue Candidate (by rejecting the Red Candidate) is primarily
determined by relative candidate desirability (Figure 4b, right panel), mirroring choices in
settings with full turnout (Figure 4b, left panel).

4. As rightly argued by the authors, this study's implications "across psychology and political
science are many, promising, and hard to reject". However, I found very little discussion of this
crucial point. Especially when it comes to political science, I would like a deeper reflection of the
relative pros and cons of rejection-based electoral processes. Normative theories of democracy
are widely based on selection, rather than rejection, when it comes to the relationship between
voters and elected in a representative democracy. Hence, an alteration of the basic premise of
voting would carry enormous implications. Negative campaigning (only briefly touched in the
discussion) could actually increase and becomes the normality of campaigning (also considering
that out-party hate has turned stronger than in-party love across established democracies
worldwide). This could in turn affect patterns of affective polarization in the electorate, with
possible spill-over effects in terms of political intolerance and even acceptance of political
violence. In my view, the authors should take these concerns seriously. As a matter of fact,
turnout may decline in rejection-based elections. But at what cost for the democratic process?
In what ways would this change affect the socio-political composition of abstainers?

We completely agree with the reviewer's insights on our work's implications for political science.
As we unpacked earlier (Re: 1 part B), our findings show that rejection-based voting could lead
to a more representative election outcome (pros), yet there is also potential for a side effect of
enhanced negativity in the political environment (cons). We've included the discussions of this
implication in the updated Discussion section. Notably, beyond the consequences of changing
voting laws and procedures, which ultimately face real-world constraints, we believe our work
could have other implications for political science.

For instance, our findings suggest that poll respondents may mask their true preferences by
responding 'undecided' in cases where they prefer one candidate over another but dislike both.
We show that rejection-based polling may be an effective means of revealing the preferences of
these voters, encouraging them to indicate their true preference rather than opting to select
'undecided.' Moreover, unlike changing the structure of elections, implementing such a change

to polling is highly feasible and relatively inexpensive, and could have immediate benefits for
forecasting elections and assessing public opinion. We now elaborate on this in the Discussion:

While our findings demonstrate that choice framing can alter decisions at the individual
level when the choice itself is altered, it remains to be determined whether this extends
to other methods for framing these decisions that do not involve changing the choice at
the ballot box, such as messaging campaigns¹⁵⁻¹⁷ aimed at reframing the choice in the
individual's mind prior to the election. Separate from these potential implications for
influencing voter turnout, our findings also have direct and more immediate implications
for assessing public opinion. We show that a relatively simple alteration to how polls are
framed has the potential to reveal the preferences of voters who may be masking those
preferences because of their dislike of their options. While this may not produce a direct
change in election outcomes, it can better inform the public about the true preferences
of the electorate, which would have important downstream benefits for policy setting
and even potentially for mobilizing voters who may otherwise misestimate preferences
toward their preferred candidates or policies. Specifically, pollsters could consider
adopting rejection-framing poll questions and explore their utility in estimating voter
intentions/preferences and ultimately forecasting election outcomes (cf. Liu et al.,
2021¹⁸).

Replies to Reviewer 2:

Across two experiments the authors test whether opt out rates in a voting choice task are driven
more by indifference, not wanting to vote for either of two bad choices, or by the mismatch
between the goal of wanting to choose the best candidate versus the task of choosing between
the lesser of two unappealing options (when no good candidate is on offer). The authors
manipulated both the absolute undesirability of each of two candidates, as well as their relative
desirability (operationalized as issue alignment with the participant), across a series of 100
choices. The authors also manipulated whether participants were asked to select the preferred
or reject the dispreferred candidate (between subjects). The critical outcome variable was rates
of opting out of voting across the 100 choices.

WHEN they voted in Exp 1, participants selected the preferred and rejected the dispreferred
candidate about $\frac{3}{4}$ of the time on average. These choices increased/got faster as the desirability
gap between the candidates grew and made it easier to distinguish them.

However, participants opted out 40% of the time in the select condition. This was more likely
when the candidates were similar, and more importantly, much more likely when the absolute
desirability was low for each candidate.

Opting out rates were lower in the reject condition: 33%. Critically, they were no more likely to
opt out in the low-low ballot cases. If anything they were more likely to opt out in high-high
choices but those rates never approached the rates in low-low/select.

Exp 2 replicated 1 despite the task change which put the opt out choice at the same decision
stage as selecting/rejecting a candidate.

A simulation building on Exp 2 points to important consequences of these outcomes: opt-out
votes are less correlated with absolute candidate desirability. This matters because it means opt-
out instances won't cluster in one region (or party, in this case) along the candidate desirability
gradient (Fig 4).

Two follow up surveys with independents converged with the basic finding: fewer opt outs in
rejection than selection frames.

It's a great paper. It's beautifully written, the experiments and surveys are simple and well
designed, and the effect sizes are "huge" (as the authors note). If I was just reading this paper
without a specific journal in mind, I'd say minor revisions (see below), if that.

Two things arise, however, when thinking about this paper in the context of Nature Comm.

1. With respect to the work's theoretical implications in political science, I'm having trouble
figuring out what readers have learned here that hasn't already been discussed in the negative
voting and ranked-choice voting literatures (including the Pew data the authors cite here: [24]).

We apologize for not clarifying the novelty of our work within political science. We believe that
our work advances the field in at least three critical ways:

First, our studies are the first to experimentally test the influence of a negative voting
manipulation on real-world measures of voter preferences, using a randomized controlled
design. Past work, such as the research cited by the reviewer, had examined who is most likely
to be a negative voter and why^{7,19-22}. However, these questions are typically examined
retrospectively (e.g., based on how voters had made their decision), which results in selection
bias.

Second, our studies are among the first to elaborate on the mechanisms underpinning such
voting decisions, bridging political science research that has typically occurred at the population
level with psychological research characterizing moment-to-moment decision dynamics within
and across individuals. In doing so, we developed and validated a method that tailors voting
decisions to an individual's preferences, making it possible to study variability in voter behavior
beyond the natural statistics afforded by real-world political choices. We include the above points
in the updated Discussion section:

To the best of our knowledge, our studies are the first to experimentally test the influence
of a negative voting manipulation on real-world measures of voter preferences, using a
randomized controlled design. Past empirical work on negative voting had examined who
is most likely to be a negative voter and why^{7,19-22}. However, these questions are typically
examined retrospectively (e.g., based on how voters had made their decision), which
results in selection bias. Our studies are also among the first to elaborate on the
mechanisms underpinning such voting decisions, bridging political science research that
has typically occurred at the population level with psychological research characterizing
moment-to-moment decision dynamics within and across individuals. In doing so, we
developed and validated a method that tailors voting decisions to an individual's
preferences, making it possible to study variability in voter behavior beyond the natural
statistics afforded by real-world political choices.

Finally, our work also has implications for normative theories regarding negative voting^{6,7} and
broader theories of representations^{8,9}. For instance, Kang (2010)⁶ theorized that negative voting
would favor a centrist candidate (i.e., less polarizing) over an extremist candidate (i.e., more
polarizing). Our work brings empirical evidence and a mechanistic framework directly to bear on

such predictions, including by simulating population-level election outcomes that could result
from different distributions of voters with varying candidate preferences. We now also elaborate
on this point in the Discussion section:

Our work also has implications for normative theories regarding negative voting^{6,7} and
broader theories of representations^{8,9}. For instance, Kang (2010)⁶ theorized that negative
voting would favor a centrist candidate (i.e., less polarizing) over an extremist candidate
(i.e., more polarizing). Our work brings empirical evidence and a mechanistic framework
directly to bear on such predictions, including by simulating population-level election
outcomes that could result from different distributions of voters with varying candidate
preferences.

2. With respect to the work's theoretical implications in decision-making, I'm having trouble
convincing myself that these findings constitute more than an application of this research group's
previous elegant work on goal congruency in value.

The reviewer is right to point out that this research takes inspiration from our earlier research on
goal congruency, and we can see how it might seem like a small jump from that work to the
current work. This is our fault in failing to better describe the gap that exists between these two
lines of research, and why this collectively represents a major leap.

First, prior to ours, there had been no studies of within-individual variability in decision dynamics
underlying voting decisions involving candidates that vary in their policy positions. Our research
therefore constitutes the first evidence that such decisions share properties with past research
in the decision sciences (e.g., on choices between gambles and consumer goods), including our
own past finding of goal congruency effects on choice behavior (e.g., option value effects on
choice RT).

Second, prior to ours, there had been no studies of the effects of goal congruency on the
influence of option value on decision avoidance, in *any* choice setting. Demonstrating that this
was the case for opt-out behavior in voting choices was therefore not trivial, in part because it
required validating basic predictions about choice dynamics in such choices and required that
these predictions carry forward to a previously untested decision variable (opting out).

We now clarify both of these advances in the updated Introduction section:

We built our work on recent theoretical and experimental findings in decision science,
which show that participants weigh information about their options (e.g., consumer
goods) differently depending on their choice goal¹⁻⁴. When facing multiple options they
like, participants choose faster when their goal is to select the best option than the worst
one. When facing multiple options they dislike, they are faster to choose the worst one

than the best one. These findings have been accounted for by models that characterize
decision-making as a process of evidence accumulation towards a decision threshold,
whereby evidence congruent with one's decision goal (choose worst vs. best) reaches
threshold fastest³⁻⁵.

Our current work builds on these previous findings in two important ways. First, by testing
whether goal-driven reversals in choice dynamics generalize from consumer choice to
voting decisions (e.g., resulting in faster choices when selecting the better of two good
candidates or choosing the worse of two bad candidates). Second, by testing, for the first
time, whether these distinct choice goals also invert patterns of decision avoidance (i.e.,
whether to choose or opt-out) depending on the quality of one's options. If decisions
about whether to choose are driven by goal congruency in the same way as decisions
about which option to choose, then decision-makers would be expected to opt-out more
or less depending on whether the quality of their option set aligns with their choice goal.
In the context of voting decisions, this would predict participants opting not to vote when
being asked to choose the better of two candidates they dislike, but not when asked to
reject the worse of the two.

And in the Discussion section:

Broadening out from their particular implications for voting decisions, our findings build
on past work examining the effect of goal congruency on everyday choices, which showed
that goal incongruency (selection among bad options or rejection among good options)
leads to longer decision time and lower decisional confidence^{18-20,34,35}. These findings
have been shown to reflect a basic property of decision-making – decisions about which
option is best accumulate to a decision boundary (i.e., the point where a person feels
ready to make a choice) more readily when choosing among good options; conversely,
decisions about which option is worst accumulate to that decision boundary more readily
when choosing between bad options¹⁸⁻²⁰. Importantly, we provide the first evidence
linking goal incongruency to decision-avoidance behavior³⁶⁻³⁹. By showing that
unattractive options do not necessarily increase voter opt-outs, our work provides new
insights into research on decision avoidance in broader contexts – from voting and polling
for public opinion to deciding how to word an e-mail. Implications across psychology and
political science are many, promising, and hard to reject.

3. As for practical applications: in a vacuum the work has implications for how we run elections.
But the reality is that changing voting laws and procedures in the U.S. (the survey case here) is

incredibly difficult. Unless the authors are thinking of testing these ideas in other democracies,
it's hard to see what the practical implications of this work are, just given realistic constraints.

We apologize for also failing to clarify the potential implications of this work. The reviewer is right
to point out that elections are unlikely to change from being selection-based to rejection-based.
However, similar ends can potentially be achieved by altering how individuals frame their votes
internally when going to the ballot box (i.e., as a vote against the lesser candidate rather than a
vote for the better one). In this sense, our findings can directly inform efforts at campaign
messaging and voter mobilization, grounding these efforts in psychological mechanisms and
quantitative simulations that can help target their efforts appropriately.

Our work also has direct implications for forecasting elections and assessing public opinion. We
show that a relatively simple alteration to how polls are framed has the potential to reveal the
preferences of voters who may be masking those preferences because of their dislike of their
options. While this may not produce a direct change in election outcomes, it can better inform
the public about the true preferences of the electorate, which would have important
downstream benefits for policy setting and even potentially for mobilizing voters who may
otherwise misestimate preferences toward their preferred candidates/policies.

While the potential efficacy of each of these interventions remains to be determined, it bears
emphasizing that the stakes are substantial. For example, within the American context, the last
presidential election was determined by roughly 2-3% of all eligible swing state voters.
Interventions that better reveal preferences and/or mobilize voters by even a small margin can
therefore have an outsize impact.

We now summarize the above points in the updated Discussion section:

While our findings demonstrate that choice framing can alter decisions at the individual
level when the choice itself is altered, it remains to be determined whether this extends
to other methods for framing these decisions that do not involve changing the choice at
the ballot box, such as messaging campaigns¹⁵⁻¹⁷ aimed at reframing the choice in the
individual's mind prior to the election. Separate from these potential implications for
influencing voter turnout, our findings also have direct and more immediate implications
for assessing public opinion. We show that a relatively simple alteration to how polls are
framed has the potential to reveal the preferences of voters who may be masking those
preferences because of their dislike of their options. While this may not produce a direct
change in election outcomes, it can better inform the public about the true preferences
of the electorate, which would have important downstream benefits for policy setting
and even potentially for mobilizing voters who may otherwise misestimate preferences
toward their preferred candidates or policies. Specifically, pollsters could consider

adopting rejection-framing poll questions and explore their utility in estimating voter
intentions/preferences and ultimately forecasting election outcomes (cf. Liu et al.,
2021¹⁸).

4. More specific questions: Why restrict the study participants to independents who make up the
smallest group of voters in the electorate relative to the two parties? Unless I missed something,
the authors didn't apply that same restriction to the experiment samples. Related, how was
independent defined? Only those who answered 0 on the Jenke & Huettel scale?

We apologize for not better clarifying each of these points.

For Studies 3-4, we defined Independents based on each participant's self-reported political
affiliations on Prolific (Democrat, Republican, Independent, Other, None). We did not use
responses to the Jenke & Huettel scale to identify these groupings.

To the question of why we restricted our sample to Independents, first we should clarify that
Independents are actually around 34% of the voting population, reflecting the second largest
group of voters in the 2024 presidential election and by far the fastest growing subset of the
electorate (in large part because younger voters are more likely to affiliate as Independent than
as Democrat or Republican²³).

More importantly, the reason for restricting Studies 3-4 to Independents was that our
simulations from Studies 1-2 (Figure 4) suggested that we would expect selection vs. rejection
framing to have the largest impact on opt-out behavior for voters who experience their candidate
choice as a lose-lose. Based on contemporaneous and historical polling data, we predicted that
self-described Independent voters would meet these criteria. To the extent that many Democrats
and Republicans also experienced their choice as a lose-lose, we would expect these groups to
show similar effects, but we would expect these effects to be weaker overall, given that many in
each of these parties viewed their choice as a win-lose. To maximize our power to detect our
effect of interest, we therefore chose to preregister and test our hypothesis within groups of
Independent voters. We now clarify this point in our revised manuscript:

Based on our simulations from Studies 1 and 2 (Figure 4), we expected the selection vs.
rejection framing to have the largest impact on opt-out behavior for those voters who
tend to experience their candidate choice in the US elections as a lose-lose. Therefore,
for both surveys, we recruited a thousand US participants on Prolific who self-identified
as Independents — a group that during this period was typically most likely to hold
negative views of both Democratic and Republican candidates²² — to respond to a poll
for the upcoming elections (final N = 967 for Study 3 and 976 for Study 4; see Methods
for inclusion and exclusion criteria).

The reason we didn't use a similar restriction in Studies 1-2 is that those studies were designed
to synthesize a range of candidates tailor-made to a participant's policy preferences, irrespective
of their party affiliation. Thus, each participant was seeing lose-lose and win-win choices (and so
on), but what constituted a lose-lose or a win-win choice varied across participants. The real
world doesn't afford these same controls, so we had to work backwards from the candidates that
were being offered (Trump/Harris or Trump/Biden) and identify participants for whom this was
most likely to be a lose-lose choice and therefore most likely to reveal a change in opt-out
behavior under selection vs. rejection frames (in the manner described above).

Replies to Reviewer 3:

This is an interesting paper which evaluates the extent to which survey respondents' willingness
to express ranked preferences over political candidates varies as a function of whether they are
asked to make positive selections (e.g., which of these candidates is better) or negative selections
(e.g., which of these candidates is worse). Fielding an interesting experimental design to two
small samples of respondents, the authors demonstrate that rejection-based choice questions
result in lower levels of abstention in hypothetical voting decisions than questions based on
acceptance-based wording. They also demonstrate that the degree to which these types of
question lead to opt out vary in predictable ways with a priori measurements of candidate quality
(such that opt out rates are higher under acceptance-based selection when both candidates are
low quality, and opt out rates are higher under rejection-based selection when both candidates
are high quality). In a second set of experiments, they then demonstrate that these dynamics
hold in a real-world setting for independent voters considering voting decisions in the context of
the US presidential election.

The paper is well-written and highlights an interesting pattern of responses in the context of
political surveys. However, I have some concerns about the study design and interpretation
which I think need to be addressed if the authors are offered the opportunity to resubmit the
manuscript.

1. My primary concern with this paper is that the first two experiments rely on very small and
unrepresentative samples. Study 1 has $N = 91$, while Study 2 has $N = 100$, both conducted via
Prolific. The inferential dangers of using small samples of this nature are well known, not just in
terms of the potential for imbalance on unobserved covariates but also in their propensity to
produce both type S and type M errors (e.g.,
<https://journals.sagepub.com/doi/10.1177/1745691614551642>).

Related to this point, I am concerned about the power of the design. I wanted to know whether
the authors had pre-registered this design (my impression is that they pre-registered the central
expectations for experiments 3 and 4, but not the expectations or the analysis strategy for
experiments 1 and 2). If they did pre-register the experiment, did they conduct a power analysis
for these designs? The effects they report are very large but given that small sample studies are
more like to suffer from both sign and magnitude bias, I think it is important to establish the
extent to which the authors had been anticipating effects of this magnitude in advance of running
the study.

We appreciate the reviewer's request that we provide additional information regarding power
for Studies 1-2, and understand the concern that sample sizes of 80-100 may not always be
sufficient for reliably identifying effects of interest in psychological experiments. We now report

on several additional analyses that we hope clarify that these samples were more than sufficient
 for capturing our effects of interest, particularly given that these two studies also served as near-
 replications of one another in terms of design (with the only difference being whether the ‘no
 vote’ option was offered coincident with or prior to the candidate choice). As is shown in the
 figure below (where each point reflects a single participant), the effect size for our key framing
 effect comparing the rejection group to the selection group (yellow vs. purple) is quite large for
 both Studies 1 and 2, requiring a sample size of $N > 46$ to achieve a power of at least 0.8
 (calculated by R package: simr). Both studies had N s of at least 82, with projected power
 saturating to 1 (CI = [0.996, 1] for both studies)¹.

 The reviewer raised additional concerns about susceptibility to sign error and magnitude error.
 To test for this, we followed procedures recommended by the citation above²⁴ to calculate these
 error rates. Since our experiments were among the first utilizing dense sampling of within-subject
 measurements to study voter decision-making, they lack a past basis for us to estimate the true
 effect sizes. In lieu of these a priori estimates, we can examine effect sizes from four studies of
 decision avoidance that we have since run, building off of the current ones. The findings above
 replicate across every single one of them with varying sample sizes (N s = 52-106) (see figure
 below). Using the effect sizes we observe in those studies (betas = 0.60-1.00) as plausible values
 of the true effect sizes² to calculate the sign error and magnitude error of our original studies, we
 found that our studies are immune to both concerns. For both studies, all the sign error rates are
 less than 10^{-6} , and the magnitude errors are less than 1.1, comparable to typical public health
 studies²⁴. These findings are in preparation for a separate manuscript (focused on characterizing

¹ Note that each of these power analyses was performed by using the observed effect size from one study (e.g., Study 1) to estimate the sample size required for the other (e.g., Study 2), to avoid issues of circularity.

² Note that the opt-out option in these follow-up studies involved deferral of a decision rather than avoiding it completely (as in our voter choice studies), so these effect sizes likely provide conservative estimates of what would be expected in further studies of decision avoidance using voter choice.

the finer-scale temporal dynamics of these decisions), but if the reviewer and editor prefer, we
 can also discuss this work in our paper to underscore the reliability of the findings.

Overall Desirability Effect on Decision Avoidance: Across Different Choice Domains and Stimuli

Collectively, these analyses suggest that Studies 1 and 2 were each, on their own, well-powered
 to identify the effects of interest. To assuage the broader concerns further, we will add two
 points. First, even if we assume no a priori basis for an estimated effect size but instead had
 predicted a medium effect size (beta = 0.5, smaller than any of those observed above), both our
 studies would still have achieved a power of at least 0.86 (Study 1: power = 0.87, CI = [0.85, 0.89];
 Study 2: power = 0.91, CI = [0.89, 0.92]). Second, it's worth noting that the moderate sample sizes
 needed for Studies 1-2 are consistent with past decision-making experiments that utilize dense
 sampling of within-subject measurements³⁻⁵, in contrast to studies that utilize between-subject
 designs with few within-subject measurements and often require much larger samples.

We've included the above points in our updated Methods section:

To examine the robustness of our findings in Studies 1 and 2, we ran power analyses on
 our core effect of interest, that is, the significant interaction between the choice goals
 and overall desirability on the probability of opt-out. Using simulation-based power

analysis (R package: *simr*²⁵), we find that both studies require a sample size of $N > 46$ to
achieve a power of at least 0.8; both studies had N s of at least 82, with projected power
saturating to 1 (CI = [0.996, 1] for both studies). To be conservative, we set the estimate
of actual effect size to be 0.5, way below the observed values (Study 1: $\beta = 1.71$; Study
2: $\beta = 1.54$), and find that both studies still achieved a power of at least 0.86 (Study 1:
power = 0.868, CI = [0.845, 0.888]; Study 2: power = 0.907, CI = [0.887, 0.924]). In addition,
we calculated the sign error rates and magnitude errors²⁴ using the same conservative
estimate, and confirmed that our study was immune to both kinds of error (sign error
rates $< 10^{-6}$, magnitude errors < 1.1 , comparable to typical public health studies²⁴).

In addition, the descriptive statistics in the appendix suggest that treatment and control groups
in both of the first two studies are not well-balanced in terms of key demographic variables such
as gender and parental education. This provokes the concern that there may also be other,
unobserved differences between treatment and control groups that are driving differences in
observed outcomes, rather than the rejection/acceptance prompt condition itself. Ultimately,
my impression is that very small online surveys of the sort reported here are generally not
sufficiently robust to warrant publication in Nature journals because of the inferential problems
associated with them, but I am open to being convinced otherwise. Given these limitations, I
think that -- at the very least -- the authors need to explicitly acknowledge the risks posed by
small, imbalanced samples and discuss how these limitations might affect the interpretation of
their results.

We hope that the analyses above help address the reviewer's central concerns about whether
our samples are large enough to draw inferences about whether selection-based and rejection-
based voting differ in the influence of overall candidate desirability on opt-out behavior.

The reviewer's final concern relates to differences in sample characteristics (e.g., demographics).
We address the broader concern about generalizability below, but with respect to the specific
question of sample differences in Studies 1-2 we have now performed a comprehensive series of
analyses to examine the extent to which potential differences across our samples contributed to
our key selection vs. rejection effect (figures above). We find that our effects of interest are
robust to controlling for a wide range of individual-level variables collected across our studies,
including measures of demographics (e.g., sex, race) and political affiliation. These analyses are
now reported in Supplementary Text B and Extended Data Table 23.

2. On representativeness: Unless I have misunderstood, I do not believe that these samples are
collected such that they are representative of the broader US population and nor do the authors
appear to use survey weighting approaches in any of their analyses. These issues therefore raise

concerns about the subsequent generalizability of the findings presented by the authors to the
ultimate population(s) of interest. Again, I think this needs to be explicitly addressed in the
manuscript.

The reviewer is right to point out that we cannot guarantee that the make-up of our samples is
representative of the US population, and this constrains any inferences we would seek to make
about how our findings generalize to this population. However, we don't think this is an issue for
the inferences we seek to draw from either of our two sets of studies.

For Studies 1-2, the question we are asking is whether voting decisions made by the average
person are influenced by overall candidate desirability, and whether the influence of candidate
desirability on their propensity to opt-out is modulated by choice framing (selection vs.
rejection). The approach we took to addressing this was to collect repeated measures (100
choices) from a given participant as we varied choice properties in a manner tailored to their own
preferences (i.e., policy positions). This is comparable to past research in the decision sciences in
which researchers vary the properties of options (e.g., tastiness and healthiness of foods) as
participants chose between them. While those studies cannot guarantee that the decision
patterns they observe will generalize to all subgroups of the population (e.g., individuals with
anorexia), they can provide strong evidence for a common set of decision mechanisms across the
population (see examples in other forms of value-based decisions^{26,27}).

For Studies 3-4, the question we are asking is whether someone who is inclined to respond
'undecided' to a poll of real-world candidates under selection-based framing will, on average, be
less likely to do so under a rejection-based framing. As in the case above, our inferences don't
depend on the population we are generalizing to, but rather only on the assumption that random
sampling will guarantee that our treatments (selection-based vs. rejection-based polls) are
representative of the same subset of individuals for both studies. Having said that, we believe
further studies are warranted to verify the generalizability of our findings to any population of
interest (e.g., Independents in the general population) by appropriate approaches such as
representative sampling or survey weighting²⁸.

We have incorporated the above points regarding the generalizability of our findings into the
updated Discussion section:

One limitation of our work is that our samples were not explicitly designed to be
representative of the US population in all respects, and this constrains our ability to
generalize to this population. Instead, our studies enable us to make strong inferences
about the general decision mechanisms that underpin decisions to vote (including the
critical interaction between candidate desirability and choice frame), and about the role
these mechanisms play in decisions to reveal one's preference when eligible voters are

polled about actual political candidates. Nevertheless, it will be important for future work
to test the extent to which our findings generalize to all cross-sections of the American
population, as well as to other countries, using approaches such as representative
sampling and/or survey weighting²⁸.

3. I didn't understand the authors' decision to include only independent voters in the third and
fourth study. By focusing attention on this subset of the American electorate, the authors
unnecessarily restrict the external validity of their study. In addition, I am concerned that it is for
this group of voters for whom we would expect to see the biggest differences between the
acceptance-based and rejection-based choice conditions (because committed Democrats and
Republicans are more likely to hold fixed attitudes over the Trump/Biden candidacies). In
essence, then, the differences reported in figure 5 are therefore likely to be upper bounds on the
size of the effects the authors are interested in. I think this needs more explanation and
clarification in the paper.

We apologize for not better clarifying our decision to include only Independents in our third and
fourth studies. As we now explain in the revised manuscript, our choice to focus on Independent
voters in Studies 3-4 was indeed because our simulations from Studies 1-2 (Figure 4) suggested
that we would expect selection vs. rejection framing to have the largest impact on opt-out
behavior for voters who experience their candidate choice as a lose-lose. Based on
contemporaneous and historical polling data, we predicted that self-described Independent
voters would meet these criteria. We now clarify this point in our revised manuscript:

Based on our simulations from Studies 1 and 2 (Figure 4), we expected the selection vs.
rejection framing to have the largest impact on opt-out behavior for those voters who
tend to experience their candidate choice in the US elections as a lose-lose. Therefore,
for both surveys, we recruited a thousand US participants on Prolific who self-identified
as Independents — a group that during this period was typically most likely to hold
negative views of both Democratic and Republican candidates²² — to respond to a poll
for the upcoming elections (final N = 967 for Study 3 and 976 for Study 4; see Methods
for inclusion and exclusion criteria).

As the reviewer points out, and as we have now clarified in our revisions, this suggests that the
effect sizes we see in Studies 3-4 are likely to be closer to an upper rather than lower bound on
the real-world effects of such a manipulation (though it is theoretically possible to find an even
stronger effect by refining our inclusion criteria further to only include those individuals who
experience the decision as a lose-lose). However, we see this by and large as a feature rather
than a bug of our approach:

First, our approach is directly guided by theory, with Studies 3-4 testing (preregistered)
predictions stemming from model-based simulations derived from Studies 1-2. To the extent our
findings in Independent voters represent an upper-bound on theorized effects, it is because the
theory predicts that this framing manipulation will matter most for voters who view their choice
as lose-lose rather than those who feel very positive about at least one of the two candidates.

Second, the target population of Independent voters is substantial, representing around 34% of
the voting population, reflecting the second largest group of voters in the 2024 presidential
election and by far the fastest growing subset of the electorate²³. Even if the effects we see in
Studies 3-4 only generalized to Independent voters (and/or more specifically “double-haters”)
and no other group, they would still be of tremendous practical importance to practitioners, who
often focus their efforts disproportionately on subsets of the population like these.

Third, it’s worth underscoring that the effects we see in Studies 3-4 are quite large (36-40%
change in sampled Independent voters reporting themselves as being ‘undecided’). For
reference, the most recent election was determined by roughly 2-3% of all eligible swing state
voters. There are of course many reasons not to draw too straight a line between one of these
estimates and the other, but this is simply to note that even if our findings represent an upper-
bound within a subset of the population, the stakes and margins are such that our findings can
be seen as having significant practical implications.

4. I also think the authors need to do more to explain the normative importance of distinguishing
between the "indifference" and "alienation" explanations for non-participation which is central
to their framing of the paper. They suggest that -- when alienation is the motivating factor behind
non-response -- this is important because "the failure to reveal their preferences can be highly
consequential". But in a voting context, in which voters are deciding between two options that
they dislike, it is not clear that it *is* consequential, as in either case we lack the information
from voters about their relative preferences over candidates. I wondered therefore whether the
authors have something more particular in mind here, which would be worth describing in more
detail in the paper.

We apologize for not clarifying enough the importance of distinguishing between indifference
and alienation in voter opt-outs. We define indifferent non-voters as those who prefer the two
candidates equally. Getting these non-voters out to vote wouldn’t change the election outcome
because both candidates would end up yielding the same number of additional votes. Alienated
non-voters, on the other hand, are those who dislike both candidates but dislike one more than
the other. That is, these non-voters still have a preference, such that getting them out to vote

can have a material impact on the election outcome by shifting the vote count in favor of their
preferred (less disliked) candidate.

The consequential nature of these votes is demonstrated by our simulations in Figure 4, where
we show that an election outcome can change on the basis of which party's voters feel alienated
and decide not to turn out as a result (even if their relative preference for the two candidates is
equal and opposite to the competing party). If we were to simulate this differential turnout for
two parties that are on average indifferent about the candidates, the election outcome would
remain the same on average, no matter how many voters turn out.

We have revised our manuscript to better clarify this distinction:

One reason why people might opt out of indicating their preference in an election is if
they would be equally satisfied with any of the outcomes (e.g., with either of two
candidates winning). To the extent opt-out behavior is purely motivated by indifference,
it can be viewed as having a negligible impact on voter representation since withholding
a vote will, on the net, have a similar impact as choosing based on equal preference (e.g.,
flipping a coin). For instance, if all non-voters felt the same way about Candidates A and
B, then getting these indifferent voters out to vote wouldn't change the election outcome
because both candidates would end up yielding the same number of additional votes.

Another reason why voters might opt out of voting is if they are unhappy with their
options. Unlike an indifference-based account, this alienation-based account proposes
that people may indeed have a preference between the two candidates but feel that this
is a "lose-lose" choice. To the extent voters opt out based on alienation rather than
indifference, the failure to reveal their preferences can be highly consequential. For
instance, a group of voters may choose not to vote because they dislike both Candidate
A and Candidate B, but dislike B more than A (i.e., have a preference for A over B). Unlike
for indifferent non-voters, getting these alienated non-voters to vote can have a material
impact on the election outcome by shifting the vote count in favor of their preferred (less
disliked) candidate (e.g., Candidate A in this case).

5. In general I found the simulation analysis presented in the paper underwhelming. The authors
use this analysis to show that reducing levels of abstention by asking rejection-based selection
questions can produce election outcomes that are more representative of average voter
preferences. However, this seems almost trivially true, as it follows directly from the fact that
there are substantial differences in abstention across the experimental conditions. There is an
enormous literature on the effects of non-random turnout variation on election outcomes, and
the simple simulation that the authors employ doesn't really add any new intuition or insight, I

didn't think. What do the authors think they are contributing through this analysis that is not
obvious from the original presentation of their experiments? If the answer to that question is
"not very much", then I'd suggest dropping the simulation analysis from the paper.

We appreciate the reviewer's suggestion. We believe our simulation analysis conveys several
messages that will be non-trivial to most readers. First, as we clarified above, the distinction
between alienated and indifferent non-voters is critical, and our simulated election showed how
alienated non-voters will specifically not be represented in the typical, selection-based elections.
Furthermore, we demonstrated that the effect of rejection framing on reducing opt-out behavior
was selective to alienated non-voters. Third, our simulated elections are consistent with a key
prediction by Kang's (2010)⁶ normative account of negative voting that a centrist candidate (i.e.,
less polarizing; the Blue Candidate in the simulation setup) would be favored over an extremist
candidate (i.e., more polarizing; the Red Candidate in the simulation setup) under rejection-
based voting. This prediction is less evident from the presentation of the experiments. We now
include this connection to previous work in the revised manuscript. Finally, we see the simulation
as bridging individual-level voter behavior and population-level poll/election outcomes, making
it comparable to real-world poll/election data.

With that said, we acknowledge that a sufficiently savvy reader may be able to extrapolate all of
the above from the statistics we present prior. However, our experience in presenting these
findings to scholars from varying research backgrounds suggests to us that including these
simulations and the associated visualizations is a net positive for driving home the intuitions
above. We have therefore opted to retain this section of the paper, but are open to feedback
from the reviewer about how to further clarify its relevance.

6. It would be helpful for the authors to link their findings to the literatures on political polling
and election forecasting. For instance, one of the interesting areas of application for their
rejection-based question form would be in prospective polling of vote intentions. Does asking
these types of questions help to improve election forecasting accuracy? If so, that would be a
practically useful contribution of the paper that is not currently explored.

This is a great suggestion, and a connection we regret not having made more explicitly. We do
indeed think that our findings bear directly, on polling methodology, both in terms of the
mechanistic insights our research offers and the preliminary proof of concept seen in Studies 3-
4 for potential applications to polling per se. Collectively, these findings suggest that rejection-
based polling can at least provide complementary information to typical selection-based polls,
filling gaps currently left by poll respondents who don't commit to one of the options being
offered (e.g., 'undecided' voters). This can lead to a more complete picture of preferences across

the electorate, including to infer hidden preferences and potentially also the likelihood that a
voter may turn out and/or the extent to which that likelihood may be altered by a negative voting
frame.

We now address these points in the updated Discussion section:

While our findings demonstrate that choice framing can alter decisions at the individual
level when the choice itself is altered, it remains to be determined whether this extends
to other methods for framing these decisions that do not involve changing the choice at
the ballot box, such as messaging campaigns¹⁵⁻¹⁷ aimed at reframing the choice in the
individual's mind prior to the election. Separate from these potential implications for
influencing voter turnout, our findings also have direct and more immediate implications
for assessing public opinion. We show that a relatively simple alteration to how polls are
framed has the potential to reveal the preferences of voters who may be masking those
preferences because of their dislike of their options. While this may not produce a direct
change in election outcomes, it can better inform the public about the true preferences
of the electorate, which would have important downstream benefits for policy setting
and even potentially for mobilizing voters who may otherwise misestimate preferences
toward their preferred candidates or policies. Specifically, pollsters could consider
adopting rejection-framing poll questions and explore their utility in estimating voter
intentions/preferences and ultimately forecasting election outcomes (cf. Liu et al.,
2021¹⁸).

**References in the Response Letter**

- 1. Shafir, E. Choosing versus rejecting: Why some options are both better and worse than
others. *Memory & Cognition* **21**, 546–556 (1993).
- 2. Sokolova, T. & Krishna, A. Take It or Leave It: How Choosing versus Rejecting Alternatives
Affects Information Processing. *J Consum Res* **43**, 614–635 (2016).
- 3. Frömer, R., Dean Wolf, C. K. & Shenhav, A. Goal congruency dominates reward value in
accounting for behavioral and neural correlates of value-based decision-making. *Nat*
*Commun* **10**, 4926 (2019).
- 4. Sepulveda, P. *et al.* Visual attention modulates the integration of goal-relevant evidence
and not value. *eLife* **9**, e60705 (2020).
- 5. Leng, X., Frömer, R., Summe, T. & Shenhav, A. Mutual inclusivity improves decision-making
by smoothing out choice's competitive edge. *Nat Hum Behav* (2024) doi:10.1038/s41562-
024-02064-7.
- 6. Kang, M. Voting as Veto. *Michigan Law Review* **108**, 1221–1281 (2010).
- 7. Garzia, D. & Ferreira Da Silva, F. *Negative Voting in Comparative Perspective*. (Springer
Nature Switzerland, Cham, 2024). doi:10.1007/978-3-031-51208-7.
- 8. Urbinati, N. & Warren, M. E. The Concept of Representation in Contemporary Democratic
Theory. *Annual Review of Political Science* **11**, 387–412 (2008).
- 9. Vieira, M. B. *Reclaiming Representation: Contemporary Advances in the Theory of Political*
*Representation*. (Taylor & Francis, 2017).
- 10. Handan-Nader, C., Myers, A. C. W. & Hall, A. B. Polarization and State Legislative Elections.
- 11. Yang, V. C., Abrams, D. M., Kernell, G. & Motter, A. E. Why Are U.S. Parties So Polarized? A

- “Satisficing” Dynamical Model. *SIAM Rev.* **62**, 646–657 (2020).
- 12. Druckman, J. N. & Levendusky, M. S. What Do We Measure When We Measure Affective
Polarization? *Public Opinion Quarterly* **83**, 114–122 (2019).
- 13. Lawlor, V. M. *et al.* Dissecting the impact of depression on decision-making. *Psychological*
*Medicine* **50**, 1613–1622 (2020).
- 14. Ging-Jehli, N. R. *et al.* Cognitive Signatures of Depressive and Anhedonic Symptoms and
Affective States Using Computational Modeling and Neurocognitive Testing. *Biological*
*Psychiatry: Cognitive Neuroscience and Neuroimaging* **9**, 726–736 (2024).
- 15. Spenkuch, J. L. & Toniatti, D. Political Advertising and Election Results*. *The Quarterly*
*Journal of Economics* **133**, 1981–2036 (2018).
- 16. Aggarwal, M. *et al.* A 2 million-person, campaign-wide field experiment shows how digital
advertising affects voter turnout. *Nat Hum Behav* **7**, 332–341 (2023).
- 17. Thaler, R. H. & Sunstein, C. R. *Nudge: Improving Decisions About Health, Wealth, and*
*Happiness*. (Penguin Books, New York, NY, 2009).
- 18. Liu, Y., Ye, C., Sun, J., Jiang, Y. & Wang, H. Modeling undecided voters to forecast elections:
From bandwagon behavior and the spiral of silence perspective. *International Journal of*
*Forecasting* **37**, 461–483 (2021).
- 19. Kernell, S. Presidential Popularity and Negative Voting: An Alternative Explanation of the
Midterm Congressional Decline of the President’s Party. *American Political Science Review*
**71**, 44–66 (1977).
- 20. Gant, M. M. & Davis, D. F. Negative Voter Support in Presidential Elections. *The Western*
*Political Quarterly* **37**, 272–290 (1984).

- 21. Garzia, D. & Ferreira da Silva, F. The Electoral Consequences of Affective Polarization?
Negative Voting in the 2020 US Presidential Election. *American Politics Research* **50**, 303–
311 (2022).
- 22. Siev, J. J., Rovenpor, D. R. & Petty, R. E. Independents, not partisans, are more likely to hold
and express electoral preferences based in negativity. *Journal of Experimental Social*
*Psychology* **110**, 104538 (2024).
- 23. Salit, J. & Reilly, T. In 2024, independent voters grew their share of the vote, split their
tickets and expanded their influence. *The Conversation* [http://theconversation.com/in-](http://theconversation.com/in-2024-independent-voters-grew-their-share-of-the-vote-split-their-tickets-and-expanded-their-influence-245125)
[2024-independent-voters-grew-their-share-of-the-vote-split-their-tickets-and-expanded-](http://theconversation.com/in-2024-independent-voters-grew-their-share-of-the-vote-split-their-tickets-and-expanded-their-influence-245125)
[their-influence-245125](http://theconversation.com/in-2024-independent-voters-grew-their-share-of-the-vote-split-their-tickets-and-expanded-their-influence-245125) (2024).
- 24. Gelman, A. & Carlin, J. Beyond Power Calculations: Assessing Type S (Sign) and Type M
(Magnitude) Errors. *Perspect Psychol Sci* **9**, 641–651 (2014).
- 25. Green, P. & MacLeod, C. J. SIMR: an R package for power analysis of generalized linear
mixed models by simulation. *Methods in Ecology and Evolution* **7**, 493–498 (2016).
- 26. Collins, A. G. E. & Shenhav, A. Advances in modeling learning and decision-making in
neuroscience. *Neuropsychopharmacol.* 1–15 (2021) doi:10.1038/s41386-021-01126-y.
- 27. Frömer, R. & Shenhav, A. Filling the gaps: Cognitive control as a critical lens for
understanding mechanisms of value-based decision-making. Preprint at
<https://doi.org/10.31234/osf.io/dnvrj> (2021).
- 28. Degtiar, I. & Rose, S. A Review of Generalizability and Transportability. *Annual Review of*
*Statistics and Its Application* **10**, 501–524 (2023).

Replies to Reviewer 1:

The revised version of the manuscript has effectively addressed all the concerns raised in my review.

We are grateful to R1 for the time and effort they invested throughout this process.

Replies to Reviewer 2:

As I noted last round, I really like this paper. I still do. I also appreciate the authors' thoughtful replies to all three reviews. The revision addresses all but one of my concerns: specifically related to the generalizability of the results.

Yes, random assignment to condition with a large enough sample will address a lot of concerns about causal inference. We generally reserve concerns about the representativeness of the sample for challenges to external validity. However, based on how small the samples are in these studies, plus some recent experiences in our own research (where we have observed completely different results in politics-related experimental contrasts, holding N constant, but depending on whether we've run the study with a convenience versus representative sample), I want to second R3's first concern.

The results aren't only limited to independents, they are limited to a likely highly biased subset of them. It isn't an accident or arbitrary norm in political science that even the experiments are run with representative samples these days. To the extent that the authors want to claim that their work has implications for campaign framing and election forecasting, the evidentiary threshold is quite a bit higher.

We thank R2 for their feedback, and understand their concern about generalizability. While we have addressed R3's first concern by showing the replicability of our results in different samples across political and non-political domains, we agree that there's still a limitation regarding generalizability in our studies, which we have previously incorporated into our Discussion section (507-515 in the previous version; 441-451 in this version). To incorporate R2's remaining concern, we make the link between the possible implications and the generalizability of our work more explicit in the updated paragraph:

Having noted the potential implications of our findings for political science, it is also worth noting that a limitation of our work is that our samples were not explicitly designed to be representative of the US population in all respects, constraining our

ability to generalize to this population (though see Supplementary Text B for robustness analyses controlling for available demographics). Rather, our studies enable stronger inferences about the general decision mechanisms that underpin decisions to vote (including the critical interaction between candidate desirability and choice frame), and about the role these mechanisms play in decisions to reveal one's preference when eligible voters are polled about actual political candidates. Nevertheless, it will be important for future work to test the extent to which our findings generalize to all cross-sections of the American population, as well as to other countries, using approaches such as representative sampling and/or survey weighting.

Replies to Reviewer 3:

I appreciate the authors engaging constructively with my comments and for implementing the changes they have made in response to both my own review and that of the other reviewers. I'm happy to recommend the article is accepted on the basis of their revisions.

I've made some comments on each of the relevant revisions below, in case it is of use to the authors.

1. Power: I am convinced by the authors' response in the memo about the power of the design of their initial experiments, at least with respect to detecting the relatively large effect sizes that they focus on. While using the estimated effect sizes from the two studies to calculate power retrospectively (as the authors do in lines 444-455 of the revision memo) is not my preferred option, as the effect size from single, low-N studies is noisy and can lead to dramatic overestimates of power (see, for example, <https://pubmed.ncbi.nlm.nih.gov/29994928/>), I am more convinced by the subsequent analyses which uses the findings from additional studies on decision avoidance to inform the power analyses.

2. Pre-registration: The authors did not directly address my question about pre-registration, which does prompt me to repeat it. Were studies 1 and 2, and the associated analysis strategy pre-registered? Given the findings on power above, I do not think this is critical but it would be worth making sure it is clear in the paper (I find the description on page 15 a little vague here).

3. On imbalance: I appreciate the authors including the additional analyses in supplementary text B and am convinced by this analysis.

4. Representativeness: The text on representativeness on page 14 is helpful, thanks.

5. Including only independents: This clarification is also helpful.

6. Alienation vs indifference: The new text is really useful for illustrating the normative importance of the two accounts.

7. Simulation analysis: Fair enough. I still don't find that this analysis too much on top of the other empirical analyses but it is the authors' paper, not mine!

8. Election forecasting/polling: I think this paragraph is a helpful addition.

We thank R3 for their general appreciation of our replies. Re 2: We apologize for not better clarifying the preregistration status of our studies 1 and 2. Our studies 1 and 2, and the associated analysis strategy, were not preregistered. We now make it clearer in the Updated Methods section:

The first three studies (two non-preregistered voting tasks and one preregistered survey study, named Studies 1, 2, and 3 in the following) were approved by Brown University's Institutional Review Board under protocol 1606001529.